# NON-LINEAR ACTIVATION SOOTHES NTK CONDITIONING FOR WIDE NEURAL NETWORKS: A STUDY IN THE RELU CASE

## ABSTRACT

Non-linear activation functions are well known to improve the expressivity of neural networks, which is the main reason of their wide implementation in neural networks. In this work, we showcase a new and interesting property of certain non-linear activations, focusing on the most popular example of its kind – Rectified Linear Unit (ReLU). By comparing the cases with and without this non-linear activation, we show that the ReLU has the following effects: (a) *better data separation*, i.e., a larger angle separation for similar data in the feature space of model gradient, and (b) *better NTK conditioning*, i.e., a smaller condition number of neural tangent kernel (NTK). Furthermore, we show that the ReLU network depth (i.e., with more ReLU activation operations) further magnifies these effects. Note that, without the non-linear activation, i.e., in a linear neural network, the data separation and NTK condition number always remain the same as in the case of a linear model, regardless of the network depth. Our results imply that ReLU activation, as well as the depth of ReLU network, helps improve the worst-case convergence rate of GD, which is closely related to the NTK condition number.

## 1 INTRODUCTION

Non-linear activation functions, such as rectified linear unit (ReLU), are well known for their ability to increase the expressivity of neural networks. A non-linearly activated neural network can approximate any continuous function to arbitrary precision, as long as there are enough neurons in the hidden layers (Hornik et al., 1989; Cybenko, 1989; Hanin & Sellke, 2017), while its linear counterpart – linear neural network, which has no non-linear activation functions applied, can only represent linear functions of the input. In addition, deeper neural networks, which have more non-linearly activated layers, have exponentially greater expressivity than shallower ones (Telgarsky, 2015; Poole et al., 2016; Raghu et al., 2017; Montufar et al., 2014; Wang et al., 2018), indicating that the network depth promotes the power of non-linear activation functions.

A natural question is: *Does the non-linear activation have other beneficial effects (especially on optimization), in addition to increasing the expressivity?* Our answer is *yes*!

In this paper, we showcase a new and interesting property of certain non-linear activations, focusing on the ReLU instance: the ReLU non-linearity improves data separation in the feature space of model gradient, and helps to decrease the condition number of neural tangent kernel (NTK). We also show that the depth of the ReLU network further magnifies these effects, namely, a deeper ReLU activated neural network has a better data separation and a smaller NTK condition number, than a shallower one.

Specifically, we first show the *better separation phenomenon*, i.e., the improved data separation for similar data in the model gradient feature space. We prove that, for an infinitely wide ReLU network $f$ at its random initialization, any pair of data input vectors $\mathbf{x}$ and $\mathbf{z}$ that have similar directions (i.e., small but non-zero angle $\theta_{in}$ between $\mathbf{x}$ and $\mathbf{z}$) become more directionally separated in the model gradient space (i.e., model gradient angle $\phi$ between $\nabla f(\mathbf{x})$ and $\nabla f(\mathbf{z})$ is larger than $\theta_{in}$). We also find that deeper ReLU networks result in even better data separation, i.e., larger $\phi$.

We further show the *better NTK conditioning* property of ReLU, i.e., smaller NTK condition number. First, we prove that, as a consequence of the better data separation, the NTK condition number of a infinitely wide ReLU network is strictly smaller than that of the Gram matrix, if the dataset contains two non-degenerate samples. Moreover, as the ReLU network depth increases, the NTK condition number monotonically decreases. Then, we remove this data size assumption on two-layer ReLU networks, and prove the same better NTK conditioning, regardless of the data size as long as the dataset is not degenerated. The intuition is that, if there exists a pair of similar inputs $\mathbf{x}$ and $\mathbf{z}$ in the training set (i.e., the angle between $\mathbf{x}$ and $\mathbf{z}$ is small), which is usually the case for large datasets, then the Gram matrix and NTK of linear neural networks must have close-to-zero smallest eigenvalues, resulting in extremely large NTK condition numbers. The ReLU activation make these similar data more separated (enlarges the small angles between data), hence it helps to increase the smallest eigenvalues of NTK, which in turn leads to a smaller NTK condition number.

Note that, when the non-linear activation is absent, as in an infinitely wide linear neural network $\bar{f}$ of any finite depth, the model gradient angle $\bar{\phi}$ is always equivalent to the input angle $\theta_{in}$, and the NTK condition number $\bar{\kappa}$ also remains identical to $\kappa_0$ of the Gram matrix. With this comparison, we conclude that the better separation phenomenon, i.e., $\phi > \theta_{in}$, and the better NTK conditioning, i.e., $\kappa < \kappa_0$, observed for ReLU networks, are attributed to the non-linear activation.

We experimentally verify these findings on finite but wide neural networks. It also suggests that these results hold for finite networks.

**Condition number and optimization theory.** Recent optimization theories showed that the NTK condition number, or the smallest eigenvalue of NTK, controls the theoretical convergence rate of gradient descent algorithms on wide neural networks (Du et al., 2018; 2019; Liu et al., 2022). Combined with these theories, our findings imply that: (a), the ReLU activation function helps improving the worst-case convergence rate of gradient descent, and (b), deeper wide ReLU networks have faster convergence rate than shallower ones. Experimentally, we indeed find that deeper ReLU networks converges faster than shallower ones.

In this paper, we focus on the special case of ReLU, the most commonly used non-linear activation function. It remains theoretically an open question what are the effects of other non-linear activations on NTK conditioning and theoretical convergence rates. While it need different analysis techniques and we would like to leave it as a future work, we provide some preliminary numerical results in Appendix F. It suggests that the non-linear activation effect on the NTK conditioning can be positive (decreasing $\kappa$, as for *tanh*) or negative (increasing $\kappa$, as for *sigmoid*). It is worth to note that, in either case, a larger network depth, where more non-linear activation are operated, magnifies the effect.

**Contributions.** We summarize our contributions below. We find that:

- the ReLU non-linearity induces better separation between similar data in the feature space of model gradient. A larger depth of the ReLU network magnifies this better separation phenomenon.

- ReLU non-linearity has the effect of decreasing the condition number of the NTK matrix. A larger depth of the ReLU network further enhances this better NTK conditioning property.

- This better NTK conditioning property leads to faster convergence rate of gradient descent. We empirically verify this on various real world datasets.

The paper is organized as follow: Section 2 describes the setting and defines the key quantities and concepts, and analyzes linear neural networks as the baseline for comparison; Section 3 and 4 discuss our main results on the better separation and better conditioning of ReLU non-linear activation, respectively; Section 5 discusses the implication on theoretical convergence rates; Section 6 concludes the paper. Proofs of theorems and main corollaries can be found in the appendix.

## 1.1 RELATED WORK

Studying a specific type of non-linear activation function, especially ReLU, is a common setting in the literature. This is largely due to the fact that ReLU has emerged to be the dominant choice of activation functions in neural networks used in practice, since Nair & Hinton (2010); Krizhevsky et al. (2012). ReLU activated neural networks have received wide research attention, ranging from optimization (Li & Yuan, 2017; Du et al., 2018; Zou et al., 2020), expressivity (Hanin & Sellke, 2017;

Yarotsky, 2017; Wang et al., 2018), generalization (Zheng et al., 2019; Ji & Telgarsky, 2019; Cao & Gu, 2020), etc.

NTK and its spectrum have been extensively studied (Lee et al., 2019; Bietti & Mairal, 2019; Liu et al., 2020; Fan & Wang, 2020; Geifman et al., 2020; Xiao et al., 2020; Nguyen et al., 2021; Belfer et al., 2021; Chen & Xu, 2021), since the discovery of constant NTK for infinitely wide neural networks (Jacot et al., 2018). Velikanov & Yarotsky (2021) shows that the NTK spectrum of an infinitely wide ReLU network asymptotically exhibits a power law. Its distribution is further shown to be similar to that of Laplace kernel (Geifman et al., 2020; Chen & Xu, 2021), and can be computed (Fan & Wang, 2020). Nguyen et al. (2021) analyzed the upper and lower bounds for the smallest NTK eigenvalue in $O()$ and $\Omega()$, respectively. With the assumption of spherically uniformly distributed data where the spectrum of (elementary-wise) power of the Gram matrix becomes simplified, Murray et al. (2023), utilizing Hermite polynomials and power series expansion of NTK, provides the order of the smallest eigenvalue of the NTK of two-layer ReLU network in the infinite width limit. Under the same data setting, Basri et al. (2019) computed the NTK eigenvalues for the two-layer ReLU network. Relying on the values of off-diagonal entries of the NTK matrix in the infinite *depth* limit, another work Xiao et al. (2020) analyzed the *asymptotic* dependence of the NTK condition number on the network depth $L$ for ReLU networks, which shows a decreasing trend as $L$ increases, consistent with our result.

In contrast to prior works, we are able to distill the effect of ReLU activation function via a sharp comparison between scenarios with and without ReLU, at any finite depth without data distribution assumption. Note that, without an assumption on data distribution, NTK spectral analysis becomes much harder and many data-distribution-dependent results may not hold any more. Moreover, at finite depth, off-diagonal entries of the NTK matrix has not converged and are typically quite different from its infinite depth limit, which makes analysis even harder.

We are aware of a prior work (Arora et al., 2018) which has results of similar flavor. It shows that the depth of a linear neural network may help to accelerate optimization via an implicit pre-conditioning of gradient descent. We note that this prior work is in an orthogonal direction, as its analysis is based on the linear neural network, which is activation-free, while our work focus on the better-conditioning effect of ReLU activation function.

## 2 SETUP AND PRELIMINARIES

**Notations for general purpose.** We denote the set $\{1, 2, \cdots, n\}$ by $[n]$. We use bold lowercase letters, e.g., $\mathbf{v}$, to denote vectors, and capital letters, e.g., $A$, to denote matrices. Given a vector, $\|\cdot\|$ denotes its Euclidean norm. Inner product between two vectors is denoted by $\langle \cdot, \cdot \rangle$. Given a matrix $A$, we denote its $i$-th row by $A_{i:}$, its $j$-th column by $A_{:j}$, and its entry at $i$-th row and $j$-th column by $A_{ij}$. We also denote the expectation (over a distribution) of a variable by $\mathbb{E}[\cdot]$, and the probability of an event by $\mathbb{P}[\cdot]$. For a model $f(\mathbf{w}; \mathbf{x})$ which has parameters $\mathbf{w}$ and takes $\mathbf{x}$ as input, we use $\nabla f$ to denote its first derivative w.r.t. the parameters $\mathbf{w}$, i.e., $\nabla f := \partial f / \partial \mathbf{w}$.

**(Fully-connected) ReLU neural network.** Let $\mathbf{x} \in \mathbb{R}^d$ be the input, $m_l$ be the width (i.e., number of neurons) of the $l$-th layer, $W^{(l)} \in \mathbb{R}^{m_l \times m_{l-1}}$, $l \in [L+1]$, be the matrix of the parameters at layer $l$, and $\sigma(z) = \max\{0, z\}$ be the ReLU activation function. A (fully-connected) ReLU neural network $f$, with $L$ hidden layers, is defined as:

$$
\begin{aligned}
\alpha^{(0)}(\mathbf{x}) &= \mathbf{x} \\
\alpha^{(l)}(\mathbf{x}) &= \frac{\sqrt{2}}{\sqrt{m_l}} \sigma\left(W^{(l)} \alpha^{(l-1)}(\mathbf{x})\right), \quad \forall l \in \{1, 2, \cdots, L\}, \\
f(\mathbf{x}) &= W^{(L+1)} \alpha^{(L)}(\mathbf{x}).
\end{aligned}
\tag{1}
$$

We also denote $\tilde{\alpha}^{(l)}(\mathbf{x}) \triangleq \frac{\sqrt{2}}{\sqrt{m_l}} W^{(l)} \alpha^{(l-1)}(\mathbf{x})$. Following the NTK initialization scheme (Jacot et al., 2018), these parameters are randomly initialized i.i.d. according to the normal distribution $\mathcal{N}(0, 1)$. The scaling factor $\sqrt{2}/\sqrt{m_l}$ is introduced to normalize the hidden neurons (Du et al., 2019). We denote the collection of all the parameters by $\mathbf{w}$.

Without loss of generality, we set the layer widths as

$$m_0 = d, \ m_{L+1} = 1, \ and \ m_l = m, \ for \ l \in [L]. \tag{2}$$

and call $m$ as the network width.

**Gradient feature and neural tangent kernel (NTK).** Given a model $f$ (e.g., a neural network) with parameters $\mathbf{w}$, we consider the vector $\nabla f(\mathbf{w}; \mathbf{x})$ is the gradient feature for the input $\mathbf{x}$. The NTK $\mathcal{K}$ is defined as

$$\mathcal{K}(\mathbf{w}; \mathbf{x}_1, \mathbf{x}_2) = \langle \nabla f(\mathbf{w}; \mathbf{x}_1), \nabla f(\mathbf{w}; \mathbf{x}_2) \rangle, \tag{3}$$

where $\mathbf{x}_1$ and $\mathbf{x}_2$ are two arbitrary network inputs. For a given dataset $\mathcal{D} = \{(\mathbf{x}_i, y_i)\}_{i=1}^n$, there is a gradient feature matrix $F$ such that each row $F_{i\cdot}(\mathbf{w}) = \nabla f(\mathbf{w}; \mathbf{x}_i)$ for all $i \in [n]$. The $n \times n$ NTK matrix $K(\mathbf{w})$ is defined such that its entry $K_{ij}(\mathbf{w})$, $i, j \in [n]$, is $\mathcal{K}(\mathbf{w}; \mathbf{x}_i, \mathbf{x}_j)$. It is easy to see that the NTK matrix

$$K(\mathbf{w}) = F(\mathbf{w})F(\mathbf{w})^T. \tag{4}$$

Note that the NTK for a linear model reduces to the Gram matrix $G$.

*Infinite width limit.* Recent discovery is that, when $m$ is sufficiently large or infinite, the NTK and gradient feature becomes almost constant during training by gradient descent (Jacot et al., 2018; Liu et al., 2020). Hence, it suffices to analyze these quantities only at the network initialization, which shall extend to all the optimization procedure.

For theoretical analysis, following Jacot et al. (2018), we focus on the infinite network width limit, while let the network depth $L$ being a fixed constant. Specifically, the width of each hidden layer goes to infinity successively. This setting allows us to analyze the NTK in a cleaner way without worrying about the uncertainty arising from different random seeds of network initialization. The difference of NTKs between infinite width and finite but large width is minimal Du et al. (2018), and converge to zero Jacot et al. (2018). We use finite network width for experimental evaluations.

**Linear neural network.** For a comparison purpose, we also consider a linear neural network $\bar{f}$, which is the same as the ReLU neural network $f$ (defined above), except that the activation function is the identity function $\sigma(z) = z$ and that the scaling factor is $1/\sqrt{m}$ (we adopt the network width setting in Eq.(2)):

$$\bar{\alpha}^{(0)}(\mathbf{x}) = \mathbf{x}, \ \bar{\alpha}^{(l)}(\mathbf{x}) = \frac{1}{\sqrt{m}}W^{(l)}\bar{\alpha}^{(l-1)}(\mathbf{x}), \ \forall l \in \{1, 2, \cdots, L\}, \ \bar{f}(\mathbf{x}) = W^{(L+1)}\bar{\alpha}^{(L)}(\mathbf{x}). \tag{5}$$

**Input feature and Gram matrix.** Given a dataset $\mathcal{D} = \{(\mathbf{x}_i, y_i)\}_{i=1}^n$, we denote its (input) feature matrix by $X$, where each row $X_{i\cdot} = \mathbf{x}_i^T$. The Gram matrix is defined as $G = XX^T \in \mathbb{R}^{d \times d}$, with each $G_{ij} = \mathbf{x}_i^T \mathbf{x}_j$.

**Condition number.** The *condition number* $\kappa$ of a positive definite matrix $A$ is defined as the ratio between its maximum eigenvalue and minimum eigenvalue:

$$\kappa = \lambda_{max}(A)/\lambda_{min}(A). \tag{6}$$

**Embedding angle and model gradient angle.** For a specific input $\mathbf{x}$, we call the vector $\alpha^{(l)}(\mathbf{x})$ as the $l$-embedding of $\mathbf{x}$. We also call $\nabla f$, i.e., the derivative of model $f$ with respect to all its parameters, as the model gradient. In the following analysis, we frequently use the following concepts: *embedding angle* and *model gradient angle*.

**Definition 2.1** (embedding angle and model gradient angle). *Given two arbitrary inputs* $\mathbf{x}, \mathbf{z} \in \mathbb{R}^d$, *define the* $l$-*embedding angle,* $\theta^{(l)}(\mathbf{x}, \mathbf{z}) \triangleq \arccos\left(\frac{\langle \alpha^{(l)}(\mathbf{x}), \alpha^{(l)}(\mathbf{z}) \rangle}{\|\alpha^{(l)}(\mathbf{x})\|\|\alpha^{(l)}(\mathbf{z})\|}\right)$, *as the angle between the* $l$-*embedding vectors* $\alpha^{(l)}(\mathbf{x})$ *and* $\alpha^{(l)}(\mathbf{z})$, *and the model gradient angle,* $\phi(\mathbf{x}, \mathbf{z}) \triangleq \arccos\left(\frac{\langle \nabla f(\mathbf{x}), \nabla f(\mathbf{z}) \rangle}{\|\nabla f(\mathbf{x})\|\|\nabla f(\mathbf{z})\|}\right)$, *as the angle between the model gradient vectors* $\nabla f(\mathbf{x})$ *and* $\nabla f(\mathbf{z})$.

We also denote $\theta^{(0)}$ by $\theta_{in}$, as $\theta^{(0)}$ is just the angle between the original inputs.

In the rest of the paper, we specifically refer the NTK matrix, NTK condition number, $l$-embedding angle and model gradient angle for the ReLU neural network as $K$, $\kappa$, $\theta^{(l)}$ and $\phi$, respectively, and refer their linear neural network counterparts as $\bar{K}$, $\bar{\kappa}$, $\bar{\theta}^{(l)}$ and $\bar{\phi}$, respectively. We also denote the condition number of Gram matrix $G$ by $\kappa_0$.

## 2.1 LINEAR NEURAL NETWORK: THE BASELINE FOR COMPARISON

To distill the effect of the non-linear activation function, we need a activation-free case as the baseline for comparison. This baseline is the linear neural network $\bar{f}$, with the same width and depth as $f$.

**Theorem 2.2.** *Consider the linear neural network $\bar{f}$ as defined in Eq.(5). In the limit of infinite network width $m \to \infty$ and at network initialization $\mathbf{w}_0$, the following relations hold:*

- *for any input $\mathbf{x} \in \mathbb{R}^d$: $\|\bar{\alpha}^{(l)}(\mathbf{x})\| = \|\mathbf{x}\|$, $\forall l \in [L]$; and $\|\nabla f(\mathbf{w}_0; \mathbf{x})\| = (L+1)\|\mathbf{x}\|$.*

- *for any inputs $\mathbf{x}, \mathbf{z} \in \mathbb{R}^d$: $\bar{\theta}^{(l)}(\mathbf{x}, \mathbf{z}) = \theta_{in}(\mathbf{x}, \mathbf{z})$, $\forall l \in [L]$; and $\bar{\phi}(\mathbf{x}, \mathbf{z}) = \theta_{in}(\mathbf{x}, \mathbf{z})$.*

This theorem states that, without a non-linear activation function, both the feature embedding maps $\alpha^{(l)} : \mathbf{x} \mapsto \alpha^{(l)}(\mathbf{x})$ and the model gradient map $\nabla f : \mathbf{x} \mapsto \nabla f(\mathbf{x})$ fail to change the geometrical relationship between any data samples. For any input pairs, the embedding angles $\bar{\theta}^{(l)}$ and $\bar{\phi}$ remain the same as the input angle $\theta_{in}$. Therefore, it is not surprising that the NTK of a linear network is the same as the Gram matrix (up to a constant factor), as formally stated in the following corollary.

**Corollary 2.3** (NTK condition number of linear networks). *Consider a linear neural network $\bar{f}$ as defined in Eq.(5). In the limit of infinite network width $m \to \infty$ and at network initialization, the NTK matrix $\bar{K} = (L+1)^2 G$. Moreover, $\bar{\kappa} = \kappa_0$.*

This corollary tells that, for a linear neural network, regardless of its depth $L$, the NTK condition number $\bar{\kappa}$ is always equal to the condition number $\kappa_0$ of the Gram matrix $G$. Therefore, any non-zero deviations, $\delta\phi \triangleq \phi - \theta_{in}$ from the input angle $\theta_{in}$, and $\delta\kappa \triangleq \kappa - \kappa_0$ from the Gram condition number $\kappa_0$, observed for a non-linearly activated network $f$, should be attributed to the corresponding non-linear activation.

## 3 ReLU INDUCES BETTER DATA SEPARATION IN MODEL GRADIENT SPACE

In this section, we show that the ReLU non-linearity helps data separation in the model gradient space. Specifically, for two arbitrary inputs $\mathbf{x}$ and $\mathbf{z}$ with small $\theta_{in}(\mathbf{x}, \mathbf{z})$, we show that the model gradient angle $\phi(\mathbf{x}, \mathbf{z})$ is strictly larger than $\theta_{in}(\mathbf{x}, \mathbf{z})$, implying a better angle separation of the two data points in the model gradient space. Moreover, we show that the model gradient angle $\phi(\mathbf{x}, \mathbf{z})$ monotonically increases with the number of layers $L$, indicating that deeper network (more ReLU non-linearity) has better angle separation.

**Embedding vectors and embedding angles.** We start with investigating the relations among the $l$-embedding vectors $\alpha^{(l)}$ and the embedding angles $\theta^{(l)}$.

**Lemma 3.1.** *Consider the ReLU network $f$ defined in Eq.(1) at its initialization, and define function $g : [0, \pi] \to [0, \pi]$ as $g(z) = \arccos\left(\frac{\pi - z}{\pi}\cos z + \frac{1}{\pi}\sin z\right)$. In the infinite network width limit $m \to \infty$, for all $l \in [L]$, the following relations hold:*

- *for any input $\mathbf{x} \in \mathbb{R}^d$, $\|\alpha^{(l)}(\mathbf{x})\| = \|\mathbf{x}\|$;*

- *for any two inputs $\mathbf{x}, \mathbf{z} \in \mathbb{R}^d$, $\theta^{(l)}(\mathbf{x}, \mathbf{z}) = g\left(\theta^{(l-1)}(\mathbf{x}, \mathbf{z})\right)$. Let $g^l(\cdot)$ be the $l$-fold composition of $g(\cdot)$, then*

$$\theta^{(l)}(\mathbf{x}, \mathbf{z}) = g^l\left(\theta_{in}(\mathbf{x}, \mathbf{z})\right). \tag{7}$$

The lemma states that, during forward propagation, the $l$-embedding vectors for each input keeps unchanged in magnitude, and the embedding angles $\theta^{(l)}$ between any two inputs are governed by the closed form function $g$. Please see Appendix A for the plot of the function and detailed discussion about its properties. As a highlight, $g$ has the following property: $g$ is approximately the identity function $g(z) \approx z$ for small $z$, i.e., $z \ll 1$. This property directly implies the following theorem.

**Theorem 3.2.** *Given any inputs $\mathbf{x}, \mathbf{z}$ such that $\theta_{in}(\mathbf{x}, \mathbf{z}) = o(1)$, for each $l \in [L]$, the $l$-embedding angle $\theta^{(l)}(\mathbf{x}, \mathbf{z})$ can be expressed as*

$$\theta^{(l)}(\mathbf{x}, \mathbf{z}) = \theta_{in}(\mathbf{x}, \mathbf{z}) - \frac{l}{3\pi}(\theta_{in}(\mathbf{x}, \mathbf{z}))^2 + o\left((\theta_{in}(\mathbf{x}, \mathbf{z}))^2\right).$$

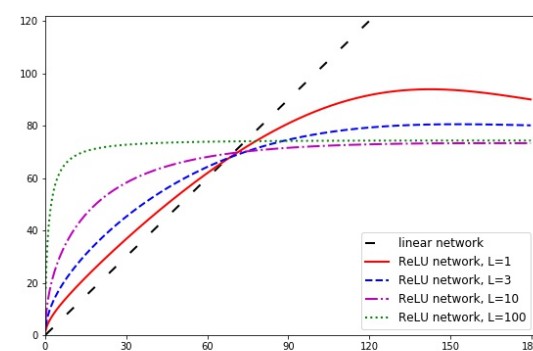

Figure 1: **Model gradient angles $\phi$ vs. input angle $\theta_{in}$ (according to Lemma 3.3).** Linear neural networks, of any depth $L$, always have $\bar\phi = \theta_{in}$, as the black dash line showed. ReLU neural networks with various depths have better data separation $\phi > \theta_{in}$ for similar data (i.e., small $\theta_{in}$). Moreover, deeper ReLU networks have better separation than shallow ones for similar data. All neural networks are infinitely wide.

We see that, at the small angle regime $\theta_{in} = o(1)$, the embedding angles $\theta^{(l)}$ at any layer $l$ is the same as the input angle $\theta_{in}$ at the lowest order. In addition, the higher order corrections are always negative making $\theta^{(l)} < \theta_{in}$. We also note that the correction term $\Delta\theta^{(l)} \triangleq \theta^{(l)} - \theta_{in}$ is linearly dependent on layer $l$ at its lowest order.

**Model gradient angle.** Now, we investigate the model gradient angle $\phi$ and its relation with the embedding angles $\theta^{(l)}$ and input angle $\theta_{in}$, for the ReLU network.

**Lemma 3.3.** *Consider the ReLU network defined in Eq.(1) with $L$ hidden layers and infinite network width $m$. Given two arbitrary inputs $\mathbf{x}$ and $\mathbf{z}$, the angle $\phi(\mathbf{x}, \mathbf{z})$ between the model gradients $\nabla f(\mathbf{x})$ and $\nabla f(\mathbf{z})$ satisfies*

$$\cos\phi(\mathbf{x}, \mathbf{z}) = \frac{1}{L+1}\sum_{l=0}^{L}\left[\cos\theta^{(l)}(\mathbf{x}, \mathbf{z})\prod_{l'=l}^{L-1}(1 - \theta^{(l')}(\mathbf{x}, \mathbf{z})/\pi)\right]. \tag{8}$$

*Moreover, $\|\nabla f(\mathbf{x})\| = (L+1)\|\mathbf{x}\|$, for any $\mathbf{x}$.*

**Better data separation with ReLU.** Comparing with Theorem 2.2 for linear neural networks, we see that the non-linear ReLU activation only affects the relative direction, but not the the magnitude, of the model gradient. Combining Lemmas 3.3 and 3.1, we get the relation between $\phi$ and the input angle $\theta_{in}$. Figure 1 plots $\phi$ as a function of $\theta_{in}$ for different network depth $L$.

The **key observation** is that: for relatively small input angles (say $\theta_{in} < 60°$), the model gradient angle $\phi$ is always greater than the input angle $\theta_{in}$. This suggests that, after the mapping $\nabla f : \mathbf{x} \mapsto \nabla f(\mathbf{x})$ from the input space to model gradient space, data inputs becomes more (directionally) separated, if they are similar in the input space (i.e., with small $\theta_{in}$). Comparing to the linear neural network case, where $\bar\phi(\mathbf{x}, \mathbf{z}) = \theta_{in}(\mathbf{x}, \mathbf{z})$ as in Theorem 2.2, we see that the ReLU non-linearity results in a better angle separation $\phi(\mathbf{x}, \mathbf{z}) > \bar\phi(\mathbf{x}, \mathbf{z})$ for similar data.

Another observation is that: deeper ReLU networks lead to larger model gradient angles, when $\theta_{in} < 60°$. This indicates that deeper ReLU networks, which has more layers of ReLU non-linear activation, makes the model gradient more separated between inputs. Note that, in the linear network case, the depth does not affect the gradient angle $\bar\phi$.

In particular, the following theorem quantifies the better data separation in the regime of small input angle $\theta_{in} = o(1)$.

**Theorem 3.4** (Better separation with ReLU). *Consider two network inputs $\mathbf{x}, \mathbf{z} \in \mathbb{R}^d$, with small input angle $\theta_{in}(\mathbf{x}, \mathbf{z}) = o(1)$, and the ReLU network defined in Eq.(1) with $L$ hidden layers and*

*infinite network width $m$. At the network initialization, the angle $\phi(\mathbf{x}, \mathbf{z})$ between the model gradients $\nabla f(\mathbf{x})$ and $\nabla f(\mathbf{z})$ satisfies*

$$\cos \phi(\mathbf{x}, \mathbf{z}) = \left(1 - \frac{L}{2\pi}\theta_{in} + o(\theta_{in})\right)\cos\theta_{in}. \tag{9}$$

Noticing the negative sign within the factor $\left(1 - \frac{L}{2\pi}\theta_{in} + o(\theta_{in})\right)$, we know that the factor is less than 1 and we obtain that: $\phi(\mathbf{x}, \mathbf{z}) > \theta_{in}(\mathbf{x}, \mathbf{z}) = \tilde{\phi}(\mathbf{x}, \mathbf{z})$. Noticing the depth $L$ dependence of this factor, we also get that: the deeper the ReLU network (i.e., larger $L$) is, the larger $\phi$ is, in the regime $\theta_{in} = o(1)$.

**Remark 3.5** (Separation in distance). *Indeed, the better angle separation discussed above implies a better separation in Euclidean distance as well. This can be easily seen by recalling from Lemma 3.3 that the model gradient mapping $\nabla f$ preserves the norm (up to a universal factor $L + 1$).*

We also point out that, Figure 1 indicates that for large input angles (say $\theta_{in} > 60°$) the model gradient angle $\phi$ is always large (greater than $60°$). Hence, non-similar data never become similar in the model gradient feature space.

## 4   ReLU INDUCES SMALLER NTK CONDITION NUMBER OF NTK

In this section, we show both theoretically and experimentally that, the ReLU non-linearity induces a decrease in the NTK condition number $\kappa$. Moreover, a ReLU network with larger depth $L$, which means more non-linear activations in operation, the NTK condition number $\kappa$ is generically smaller.

**Connection between condition number and model gradient angle.**   The smallest eigenvalue and condition number of NTK are closely related to the smallest model gradient angle $\min_{i,j\in[n]} \phi(\mathbf{x}_i, \mathbf{x}_j)$, through the gradient feature matrix $F$. Think about the case if $\phi(\mathbf{x}_i, \mathbf{x}_j) = 0$ (i.e., $\nabla f(\mathbf{x}_i)$ is parallel to $\nabla f(\mathbf{x}_j)$) for some $i, j \in [n]$, then $F$, hence NTK $K$, is not full rank and the smallest eigenvalue $\lambda_{min}(K)$ is zero, leading to an infinite condition number $\kappa$. Similarly, if $\min_{i,j\in[n]} \phi(\mathbf{x}_i, \mathbf{x}_j)$ is small, the smallest eigenvalue $\lambda_{min}(K)$ is also small, and condition number $\kappa$ is large, as stated in the following proposition (see proof in Appendix B).

**Proposition 4.1.** *Consider a $n \times n$ positive definite matrix $A = BB^T$, where matrix $B \in \mathbb{R}^{n \times d}$, with $d > n$, is of full row rank. Suppose that there exist $i, j \in [n]$ such that the angle $\phi$ between vectors $B_{i\cdot}$ and $B_{j\cdot}$ is small, i.e., $\phi \ll 1$, and that there exist constant $C > c > 0$ such that $c \leq \|B_{k\cdot}\| \leq C$ for all $k \in [n]$. Then, the smallest eigenvalue $\lambda_{min}(A) = O(\phi^2)$, and the condition number $\kappa = \Omega(1/\phi^2)$.*

Therefore, a good data angle separation in the model gradient features, i.e., $\min_{i,j\in[n]} \phi(\mathbf{x}_i, \mathbf{x}_j)$ not too small, is a necessary condition such that the condition number $\kappa$ is not too large. As is shown in the last section, the ReLU non-linearity makes the samples more separated when mapped from the input data space to the model gradient feature space. Hence, it is expected that the NTK condition number will decrease in the presence of the ReLU non-linearity.

**Smaller NTK condition number.**   Theoretically, we consider the infinite width limit. We require that the dataset is not degenerated, i.e., $\mathbf{x}_i \nparallel \mathbf{x}_j$ for all $i, j$. This is a mild and commonly used setting in the literature, see for example Du et al. (2018). We require that the first layer weights $W^{(1)}$ be trainable and fix the other layers in the following theorem. This is also a common setting in literature to simplify the analysis Du et al. (2018).

**Theorem 4.2.** *Consider the ReLU network in Eq.(38) in the limit $m \to \infty$ and at initialization. Let the first layer weights $W^{(1)}$ be trainable and fix the other layers. We compare the two scenarios: (a) the network with the ReLU activation, and (b) the network with all the ReLU activation removed. The smallest eigenvalue $\lambda_{min}(K)$ of its NTK in scenario (a) is larger than that in scenario (b): $\lambda_{min}(K_a) > \lambda_{min}(K_b)$, and the NTK condition number $\kappa$ in scenario (a) is less than that in scenario (b): $\kappa_a < \kappa_b$. Moreover, for two ReLU neural networks $f_1$ of depth $L_1$ and $f_2$ of depth $L_2$ with $L_1 > L_2$, we have $\kappa_{f_1} < \kappa_{f_2}$.*

This theorem confirms the expectation that the NTK condition number $\kappa$ should be decreased, as a consequence of the existence of the ReLU non-linearity. This theorem also shows that the depth of ReLU network enhances this better NTK conditioning.

The high-level intuition behind the proof of this theorem is that: the derivative of ReLU function, $\sigma'(z) = \mathbb{I}_{\{z \geq 0\}}$, resembles a binary gate which has *open* and *close* states. When there are ReLU implemented, the model gradient map $\nabla f : \mathbf{x} \mapsto \nabla f(x)$ increases the directional diversity of the vectors $\nabla f(x)$, thanks to the high dimension of model gradient space and the different activation patterns of the hidden layer for different samples $\mathbf{x}$. Hence, it is expected that the feature matrix $F$, as well as the NTK matrix $K$, is better conditioned.

Indeed, fixing the top layer weights is not a necessary requirement and can be removed. In our experiments in Section 4.1 where all the layers are trainable, we observe the phenomena of *better data separation* and *better NTK conditioning*. Theoretically, We consider the special case where the dataset is of size 2.

**Theorem 4.3.** *Consider a $L$-layer ReLU neural network $f$ as defined in Eq.(1) in the infinite width limit $m \to \infty$ and at initialization. We compare the NTK condition numbers $\kappa_a$ and $\kappa_b$ of the two scenarios: (a) the network with the ReLU activation, and (b) the network with all the ReLU activation removed. Consider the dataset $\mathcal{D} = \{(\mathbf{x}_1, y_1), (\mathbf{x}_2, y_2)\}$ with the input angle $\theta_{in}$ between $\mathbf{x}_1$ and $\mathbf{x}_2$ small, $\theta_{in} = o(1)$. Then, the NTK condition number $\kappa_a < \kappa_b$. Moreover, for two ReLU neural networks $f_1$ of depth $L_1$ and $f_2$ of depth $L_2$ with $L_1 > L_2$, we have $\kappa_{f_1} < \kappa_{f_2}$.*

### 4.1 EXPERIMENTAL EVIDENCE

Here, we experimentally show that better data separation and better conditioning happen in practice.

**Dataset.** We use the following datasets: synthetic dataset, MNIST (LeCun et al., 1998), FashionM-NIST (f-MNIST) (Xiao et al., 2017), SVHN (Netzer et al., 2011) and Librispeech (Panayotov et al., 2015). The synthetic data consists of 2000 samples which are randomly drawn from a 5-dimensional Gaussian distribution with zero-mean and unit variance. The MNIST, f-MNIST and SVHN datasets are image datasets where each input is an image. The Librispeech is a speech dataset including 100 hours of clean speeches. In the experiments, we use a subset of Librispeech with $50,000$ samples, and each input is a 768-dimensional vector representing a frame of speech audio and we follow (Hui & Belkin, 2020) for the feature extraction.

**Models.** For each of the datasets, we use a ReLU activated fully-connected neural network architecture to process. The ReLU network has $L$ hidden layers, and has $512$ neurons in each of its hidden layers. The ReLU network uses the NTK parameterization and initialization strategy (see (Jacot et al., 2018)). For each dataset, we vary the network depth $L$ from 0 to 10. Note that $L = 0$ corresponding to the linear model case. In addition, for comparison, we use a linear neural network, which has the same architecture with the ReLU network except the absence of activation function.

**Results.** For each dataset and given network depth $L$, we evaluate both the smallest pairwise model gradient angle $\min_{i,j \in [n]} \phi(\mathbf{x}_i, \mathbf{x}_j)$ and the NTK condition number $\kappa$, at the network initialization. We take 5 independent runs over 5 random initialization seeds, and report the average. In each run, we used a A-100 GPU to compute the NTK, which took $4 \sim 10$ hours. The results are shown in Figure 2. We compare the two scenarios of *with* and *without* the ReLU activation function. As one can easily see from the plots, a ReLU network (depth $L = 1, 2, \cdots, 10$) always have a better separation of data (i.e., larger smallest pairwise model gradient angle), and a better NTK conditioning (i.e., smaller NTK condition number), than its corresponding linear network (compare the solid line and dash line of the same color). Furthermore, the monotonically decreasing NTK condition number shows that a deeper ReLU network have a better conditioning of NTK.

## 5 OPTIMIZATION ACCELERATION

Recently studies showed strong connections between the NTK condition number and the theoretical convergence rate of gradient descent algorithms on wide neural networks (Du et al., 2018; 2019; Soltanolkotabi et al., 2018; Allen-Zhu et al., 2019; Zou et al., 2020; Oymak & Soltanolkotabi, 2020; Liu et al., 2022). In Du et al. (2018; 2019), the authors derived the worst-case convergence rates

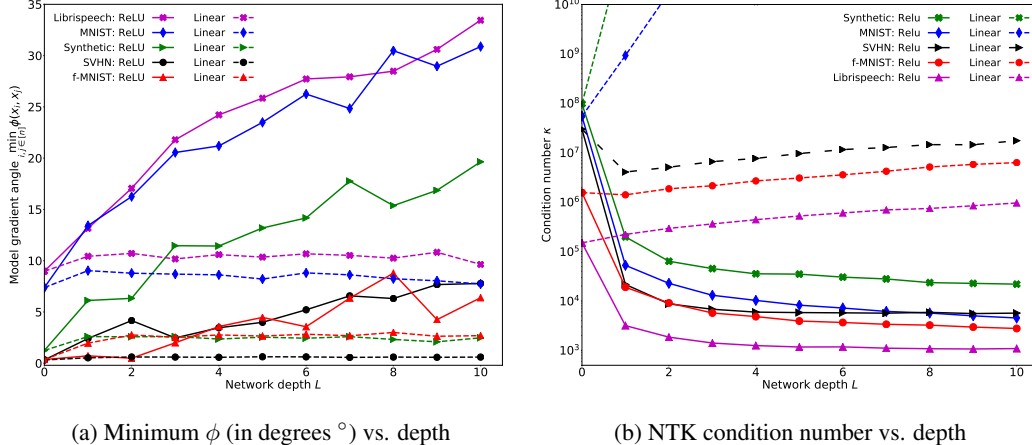

(a) Minimum $\phi$ (in degrees $^\circ$) vs. depth   (b) NTK condition number vs. depth

Figure 2: **Better separation (left) and better NTK conditioning (right) of ReLU network.** Solid lines are for ReLU networks, and dash lines are for linear networks. **Left:** ReLU network works better in separating similar data, while linear network remains similar to a linear model. **Right:** ReLU network has better conditioning of NTK than linear network and linear model. Note that $L = 0$ corresponds to the case of a linear model, and the NTK in this case is the Gram matrix.

explicitly in terms of the smallest eigenvalue of NTK $\lambda_{min}(K)$, $L(\mathbf{w}_t) \leq (1 - \eta \lambda_{min}(K)/2)^t L(\mathbf{w}_0)$, where $L$ is the square loss function and $t$ is the time stamp of the algorithm. Later on, in Liu et al. (2022), the NTK condition number is explicitly involved in the convergence rate:

$$L(\mathbf{w}_t) \leq (1 - \kappa^{-1})^t L(\mathbf{w}_0). \tag{10}$$

Although $\kappa$ is evaluated on the whole optimization path, all these theories used the fact that NTK is almost constant for wide neural networks and an evaluation at the initialization $\mathbf{w}_0$ is enough.

As a smaller NTK condition number (or larger smallest eigenvalue of NTK) implies a faster worst-case convergence rate, our findings suggest that: (a), the ReLU activation function helps improving the worst-case convergence rate of gradient descent, and (b), deeper wide ReLU networks have faster convergence rate than shallower ones.

We experimentally verify this implication. Specifically, we train the ReLU networks, with depth $L$ ranging from 1 to 10, for the datasets MNIST, f-MNIST and Librispeech. For all the training tasks, we use cross entropy loss as the objective function and use mini-batch stochastic gradient descent (SGD) of batch size 500 to optimize. For each task, we find its optimal learning rate by grid search. On MNIST and f-MNIST, we train 500 epochs, and on Librispeech, we training 2000 epochs.

The curves of training loss against epochs are shown in Figure 3. We observe that, for all these datasets, a deeper ReLU network always converges faster than shallower ones. This is consistent with the theoretical prediction that the deeper ReLU network, which has smaller NTK condition number, has faster theoretical convergence rate.

## 6 CONCLUSION AND DISCUSSIONS

In this work, we showed the beneficial effects of ReLU non-linear activation on the data separation in feature space and on the NTK conditioning. We also showed that more sequential ReLU activation operations, i.e., larger network depth, magnifies these effects. As the NTK conditioning is closely related to theoretical convergence rate of gradient descent, our findings also suggest a positive role of the ReLU activation function in optimization theories.

**Infinite depth.** In this work, we focused on the finite depth scenario which is the more interesting case from a practical point of view. Our small angle regime analysis (Theorem 3.2, 3.4 and 4.3) do not directly extend to the infinite depth case. But, as Lemma 3.3 and Figure 1 indicate, the $\phi(\theta_{in})$ function seems to converge to a step function when $L \to \infty$, which implies orthogonality between

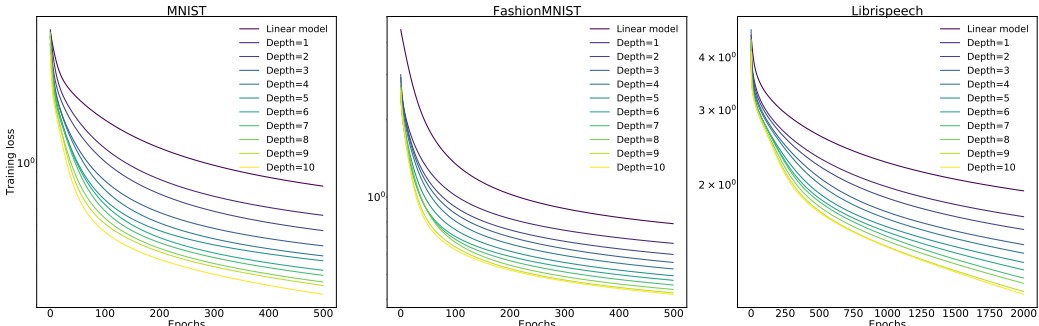

Figure 3: **Training curve of ReLU networks with different depths.** On each of these datasets, we see that deeper ReLU network always converges faster than shallower ones.

model gradient vectors, hence a NTK condition number being 1. This is consistent with the prior knowledge that NTK converges to 1 in the infinite depth limit (Radhakrishnan et al., 2023).

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

# A    PROPERTIES OF FUNCTION $g$

Recall that the function $g : [0, \pi) \to [0, \pi)$ is defined as (see Lemma 3.1)

$$g(z) = \arccos\left(\frac{\pi - z}{\pi}\cos z + \frac{1}{\pi}\sin z\right), \tag{11}$$

Figure 4 shows the plot of this function. From the plot, we can easily find the following properties.

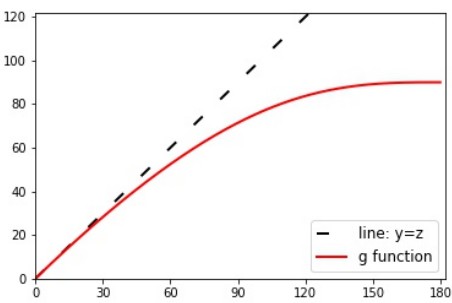

Figure 4: Curve of the function $g(\theta)$. As can be seen, $g(\theta)$ is monotonic, and is approximately the identity function $y = \theta$ in the small angle region ($\theta \ll 90°$).

**Proposition A.1** (Properties of $g$). *The function $g$ defined in Eq.(11) has the following properties:*

> *1. $g$ is a monotonically increasing function;*
>
> *2. $g(z) \leq z$, for all $z \in [0, \pi)$; and $g(z) = z$ if and only if $z = 0$;*
>
> *3. for any $z \in [0, \pi)$, the sequence $\{g^l(z)\}_{l=1}^{\infty}$ is monotonically decreasing, and has the limit $\lim_{l \to \infty} g^l(z) = 0$.*

It is worth to note that the last property of $g$ function immediately implies the collapse of embedding vectors from different inputs in the infinite depth limit $L \to \infty$. This embedding collapse has been observed in prior works Poole et al. (2016); Schoenholz et al. (2016) (although by different type of analysis) and has been widely discussed in the literature of Edge of Chaos.

**Theorem A.2.** *Consider the same ReLU neural network as in Lemma 3.1. Given any two inputs $\mathbf{x}, \mathbf{z} \in \mathbb{R}^d$, the sequence of angles between their $l$-embedding vectors, $\{\theta^{(l)}(\mathbf{x}, \mathbf{z})\}_{l=1}^{L}$, is monotonically decreasing. Moreover, in the limit of infinite depth,*

$$\lim_{L \to \infty} \theta^{(L)}(\mathbf{x}, \mathbf{z}) = 0, \tag{12}$$

*and there exists a vector $\alpha$ such that, for any input $\mathbf{x}$, the last layer $L$-embedding*

$$\alpha^{(L)}(\mathbf{x}) = \|\mathbf{x}\|\alpha. \tag{13}$$

*Proof of Proposition A.1.* **Part 1.** First, we consider the auxiliary function $\tilde{g}(z) = \frac{\pi - z}{\pi}\cos z + \frac{1}{\pi}\sin z$. We see that

$$\frac{d\tilde{g}(z)}{dz} = -\left(1 - \frac{z}{\pi}\right)\sin z \leq 0, \quad \forall z \in [0, \pi).$$

Hence, $\tilde{g}(z)$ is monotonically decreasing on $[0, \pi)$. Combining with the monotonically decreasing nature of the arccos function, we get that $g$ is monotonically increasing.

**Part 2.** It suffices to prove that $\cos z \leq \tilde{g}(z)$ and that the equality holds only at $z = 0$. For $z = 0$, it is easy to check that $\cos z = \tilde{g}(z)$, as both $z$ and $\sin z$ are zero. For $z \in (0, \pi/2)$, noting that $\tan z - z > 0$, we have

$$\tilde{g}(z) = \frac{\pi - z}{\pi}\cos z + \frac{1}{\pi}\sin z = \cos z + \frac{1}{\pi}(-z + \tan z)\cos z > \cos z. \tag{14}$$

For $z = \pi/2$, we have $\cos \pi/2 = 0 < 1/\pi = \tilde{g}(\pi/2)$. For $z \in (\pi/2, \pi)$, we have the same relation as in Eq.(14). The only differences are that, in this case, $\cos z < 0$ and $\tan z - z < 0$. Therefore, we still get $\tilde{g}(z) > \cos z$ for $z \in (\pi/2, \pi)$.

**Part 3.** From part 2, we see that $g(z) < z$ for all $z \in (0, \pi)$. Hence, for any $l$, $g^{l+1}(z) < g^l(z)$. Moreover, since $z = 0$ is the only fixed point such that $g(z) = z$, in the limit $l \to \infty$, $g^l(z) \to 0$. $\quad\square$

# B    PROOF OF PROPOSITION 4.1

*Proof.* Consider the matrix $B$ and the $n$ vectors $\mathbf{b}_k \triangleq B_{k\cdot}$, $k \in [n]$. The smallest singular value square of matrix $B$ is defined as

$$\sigma_{min}^2(B) = \min_{\mathbf{v} \neq 0} \frac{\mathbf{v}^T B B^T \mathbf{v}}{\mathbf{v}^T \mathbf{v}} = \min_{\mathbf{v} \neq 0} \frac{\|\sum_k v_k \mathbf{b}_k\|^2}{\|\mathbf{v}\|^2}.$$

Since the angle $\phi$ between $\mathbf{b}_i = B_{i\cdot}$ and $\mathbf{b}_j = B_{j\cdot}$ is small, let $\mathbf{v}'$ be the vector such that $v'_i = \|\mathbf{b}_j\|$, $v'_j = -\|\mathbf{b}_i\|$ and $v'_k = 0$ for all $k \neq i, j$. Then

$$\sigma_{min}^2(B) \leq \frac{\|\sum_k v'_k \mathbf{b}_k\|^2}{\|\mathbf{v}'\|^2} = \left\| \frac{\|\mathbf{b}_j\|}{\sqrt{\|\mathbf{b}_i\|^2 + \|\mathbf{b}_j\|^2}} \mathbf{b}_i - \frac{\|\mathbf{b}_i\|}{\sqrt{\|\mathbf{b}_i\|^2 + \|\mathbf{b}_j\|^2}} \mathbf{b}_j \right\|^2$$

$$= \frac{2\|\mathbf{b}_i\|^2 \|\mathbf{b}_j\|^2}{\|\mathbf{b}_i\|^2 + \|\mathbf{b}_j\|^2} (1 - \cos \phi)$$

$$= \frac{\|\mathbf{b}_i\|^2 \|\mathbf{b}_j\|^2}{\|\mathbf{b}_i\|^2 + \|\mathbf{b}_j\|^2} \phi^2 + O(\phi^4).$$

Since $A = BB^T$, the smallest eigenvalue $\lambda_{min}(A)$ of $A$ is the same as $\sigma_{min}^2(B)$.

On the other hand, the largest eigenvalue $\lambda_{max}(A)$ of matrix $A$ is lower bounded by $\mathrm{tr}(A)/n$. Note that the diagonal entries $A_{kk} = \|\mathbf{b}_k\|$. Hence, $c \leq \lambda_{max}(A) \leq C$. Therefore, the condition number $\kappa = \lambda_{max}(A)/\lambda_{min}(A) = \Omega(1/\phi^2)$. $\quad\square$

# C    PROOFS OF THEOREMS FOR LINEAR NEURAL NETWORK

## C.1    PROOF OF THEOREM 2.2

*Proof.* First of all, we provide a useful lemma.

**Lemma C.1.** *Consider a matrix $A \in \mathbb{R}^{m \times d}$, with each entry of $A$ is i.i.d. drawn from $\mathcal{N}(0, 1)$. In the limit of $m \to \infty$,*

$$\frac{1}{m} A^T A \to I_{d \times d}, \quad \text{in probability.} \tag{15}$$

We first consider the embedding vectors $\bar{\alpha}^{(l)}$ and the embedding angles $\bar{\theta}^{(l)}$. By definition in Eq.(5), we have, for all $l \in [L]$ and input $\mathbf{x} \in \mathbb{R}^d$,

$$\bar{\alpha}^{(l)}(\mathbf{x}) = \frac{1}{m^{l/2}} W^{(l)} W^{(l-1)} \cdots W^{(1)} \mathbf{x}. \tag{16}$$

Note that at the network initialization entries of $W^{(l)}$ are i.i.d. and follows $\mathcal{N}(0, 1)$. Hence, the inner product

$$\langle \bar{\alpha}^{(l)}(\mathbf{x}), \bar{\alpha}^{(l)}(\mathbf{z}) \rangle = \frac{1}{m^l} \mathbf{x}^T W^{(1)T} \cdots W^{(l-1)T} W^{(l)T} W^{(l)} W^{(l-1)} \cdots W^{(1)} \mathbf{z} \overset{(a)}{=} \mathbf{x}^T \mathbf{z},$$

where in step (a) we recursively applied Lemma C.1 $l$ times. Putting $\mathbf{z} = \mathbf{x}$, we get $\|\bar{\alpha}^{(l)}(\mathbf{x})\| = \|\mathbf{x}\|$, for all $l \in [L]$. By the definition of embedding angles, it is easy to check that $\bar{\theta}^{(l)}(\mathbf{x}, \mathbf{z}) = \theta_{in}(\mathbf{x}, \mathbf{z})$, for all $l \in [L]$.

Now, we consider the model gradient $\nabla \bar{f}$ and the model gradient angle $\bar{\phi}$. As we consider the model gradient only at network initialization, we don't explicitly write out the dependence on $\mathbf{w}_0$, and we write $\nabla \bar{f}(\mathbf{w}_0, \mathbf{x})$ simply as $\nabla \bar{f}(\mathbf{x})$. The model gradient $\nabla \bar{f}$ can be decomposed as

$$\nabla \bar{f}(\mathbf{x}) = (\nabla_1 \bar{f}(\mathbf{x}), \nabla_2 \bar{f}(\mathbf{x}), \cdots, \nabla_{L+1} \bar{f}(\mathbf{x})), \;\; with \; \nabla_l \bar{f}(\mathbf{x}) = \frac{\partial \bar{f}(\mathbf{x})}{\partial W^{(l)}}, \forall l \in [L+1]. \quad (17)$$

Hence, the inner product

$$\langle \nabla \bar{f}(\mathbf{x}), \nabla \bar{f}(\mathbf{z}) \rangle = \sum_{l=1}^{L+1} \langle \nabla_l \bar{f}(\mathbf{x}), \nabla_l \bar{f}(\mathbf{z}) \rangle,$$

and for all $l \in [l+1]$,

$$\langle \nabla_l \bar{f}(\mathbf{x}), \nabla_l \bar{f}(\mathbf{z}) \rangle = \langle \bar{\alpha}^{(l-1)}(\mathbf{x}), \bar{\alpha}^{(l-1)}(\mathbf{z}) \rangle \cdot \langle \prod_{l'=l+1}^{L+1} \frac{1}{\sqrt{m}} W^{(l')T}, \prod_{l'=l+1}^{L+1} \frac{1}{\sqrt{m}} W^{(l')T} \rangle \overset{(b)}{=} \mathbf{x}^T \mathbf{z}.$$

Here in step (b), we again applied Lemma C.1. Therefore,

$$\langle \nabla \bar{f}(\mathbf{x}), \nabla \bar{f}(\mathbf{z}) \rangle = (L+1) \mathbf{x}^T \mathbf{z}. \quad (18)$$

Putting $\mathbf{z} = \mathbf{x}$, we get $\|\nabla f(\mathbf{x})\| = (L+1)\|\mathbf{x}\|$. By the definition of model gradient angle, it is easy to check that $\bar{\phi}(\mathbf{x}, \mathbf{z}) = \theta_{in}(\mathbf{x}, \mathbf{z})$. $\qquad\square$

# D PROOFS OF THEOREMS FOR RELU NETWORK

## D.1 PRELIMINARY RESULTS

Before the proofs, we introduce some useful notations and lemmas. The proofs of these lemmas are deferred to Appendix E.

Given a vector $\mathbf{v} \in \mathbb{R}^p$, we define the following diagonal indicator matrix:

$$\mathbb{I}_{\{\mathbf{v} \geq 0\}} = \mathsf{diag}\left( \mathbb{I}_{\{v_1 \geq 0\}}, \mathbb{I}_{\{v_2 \geq 0\}}, \cdots, \mathbb{I}_{\{v_p \geq 0\}} \right), \quad (19)$$

with

$$\mathbb{I}_{\{v_i \geq 0\}} = \begin{cases} 1 & v_i \geq 0, \\ 0 & v_i < 0. \end{cases}$$

**Lemma D.1.** *Consider two vectors* $\mathbf{v}_1, \mathbf{v}_2 \in \mathbb{R}^p$ *and a $p$-dimensional random vector* $\mathbf{w} \sim \mathcal{N}(0, I_{p \times p})$. *Denote $\theta$ as the angle between* $\mathbf{v}_1$ *and* $\mathbf{v}_2$, *i.e.,* $\cos\theta = \frac{\langle \mathbf{v}_1, \mathbf{v}_2 \rangle}{\|\mathbf{v}_1\|\|\mathbf{v}_2\|}$. *Then, the probability*

$$\mathbb{P}[(\mathbf{w}^T \mathbf{v}_1 \geq 0) \wedge (\mathbf{w}^T \mathbf{v}_2 \geq 0)] = \frac{1}{2} - \frac{\theta}{2\pi}. \quad (20)$$

**Lemma D.2.** *Consider two arbitrary vectors* $\mathbf{v}_1, \mathbf{v}_2 \in \mathbb{R}^p$ *and a random matrix* $W \in \mathbb{R}^{q \times p}$ *with entries* $W_{ij}$ *i.i.d. drawn from* $\mathcal{N}(0,1)$. *Denote $\theta$ as the angle between* $\mathbf{v}_1$ *and* $\mathbf{v}_2$, *and define* $\mathbf{u}_1 = \frac{\sqrt{2}}{\sqrt{q}}\sigma(W\mathbf{v}_1)$ *and* $\mathbf{u}_2 = \frac{\sqrt{2}}{\sqrt{q}}\sigma(W\mathbf{v}_2)$. *Then, in the limit of* $q \to \infty$,

$$\langle \mathbf{u}_1, \mathbf{u}_2 \rangle = \frac{1}{\pi} \left( (\pi - \theta)\cos\theta + \sin\theta \right) \|\mathbf{v}_1\|\|\mathbf{v}_2\|. \quad (21)$$

**Lemma D.3.** *Consider two arbitrary vectors* $\mathbf{v}_1, \mathbf{v}_2 \in \mathbb{R}^p$ *and two random matrices* $U \in \mathbb{R}^{s \times q}$ *and* $W \in \mathbb{R}^{q \times p}$, *where all entries* $U_{ij}$, $i \in [s]$ *and* $j \in [q]$, *and* $W_{kl}$, $k \in [q]$ *and* $l \in [p]$, *are i.i.d. drawn from* $\mathcal{N}(0,1)$. *Denote $\theta$ as the angle between* $\mathbf{v}_1$ *and* $\mathbf{v}_2$, *and define matrices* $A_1 = \frac{\sqrt{2}}{\sqrt{q}} U \mathbb{I}_{\{W\mathbf{v}_1 \geq 0\}}$ *and* $A_2 = \frac{\sqrt{2}}{\sqrt{q}} U \mathbb{I}_{\{W\mathbf{v}_2 \geq 0\}}$. *Then, in the limit of* $q \to \infty$, *the matrix*

$$A_1 A_2^T = \frac{\pi - \theta}{\pi} I_{s \times s}. \quad (22)$$

**Lemma D.4.** *Consider matrix* $B = AA^T$ *with* $A \in \mathbb{R}^{n \times p}$ *and a random matrix* $W \in \mathbb{R}^{q \times p}$ *where all entries of* $W$ *are i.i.d. drawn from* $\mathcal{N}(0,1)$. *Define the tensor* $\mathbf{A}' \in \mathbb{R}^{n \times p \times q}$, *such that* $\mathbf{A}'_{ikl} := \sqrt{2} A_{ik} \mathbb{I}_{\{W_{l:} A_{i:} \geq 0\}}$. *Let* $B' \in \mathbb{R}^{n \times n}$ *be the matrix such that each entry* $B'_{ij} = \sum_{k,l} \mathbf{A}'_{ikl} \mathbf{A}'_{jkl}$. *Then, in the limit of* $q \to \infty$, *the smallest and largest eigenvalues satisfy:* $\lambda_{min}(B') > \lambda_{min}(B)$, *and* $\lambda_{max}(B') < \lambda_{max}(B)$.

## D.2  PROOF OF LEMMA 3.1

*Proof.* Consider an arbitrary layer $l \in [L]$ of the ReLU neural network $f$ at initialization. Given two arbitrary network inputs $\mathbf{x}, \mathbf{z} \in \mathbb{R}^d$, the inputs to the $l$-th layer are $\alpha^{(l-1)}(\mathbf{x}))$ and $\alpha^{(l-1)}(\mathbf{z}))$, respectively.

By definition, we have

$$\alpha^{(l)}(\mathbf{x}) = \sqrt{\frac{2}{m}}\sigma\left(W^{(l)}\alpha^{(l-1)}(\mathbf{x})\right), \ \ \alpha^{(l)}(\mathbf{z}) = \sqrt{\frac{2}{m}}\sigma\left(W^{(l)}\alpha^{(l-1)}(\mathbf{z})\right), \tag{23}$$

with entries of $W^{(l)}$ being i.i.d. drawn from $\mathcal{N}(0,1)$. Recall that, by definition, the angle between $\alpha^{(l-1)}(\mathbf{x}))$ and $\alpha^{(l-1)}(\mathbf{z}))$ is $\theta^{(l-1)}(\mathbf{x}, \mathbf{z})$. Applying Lemma D.2, we immediately have the inner product

$$\langle \alpha^{(l)}(\mathbf{z}), \alpha^{(l)}(\mathbf{x}) \rangle = \frac{1}{\pi}\left((\pi - \theta^{(l-1)}(\mathbf{x}, \mathbf{z}))\cos\theta^{(l-1)}(\mathbf{x}, \mathbf{z}) + \sin\theta^{(l-1)}(\mathbf{x}, \mathbf{z})\right)$$
$$\times \|\alpha^{(l-1)}(\mathbf{x})\|\|\alpha^{(l-1)}(\mathbf{z})\|. \tag{24}$$

In the special case of $\mathbf{x} = \mathbf{z}$, we have $\theta^{(l-1)}(\mathbf{x}, \mathbf{z}) = 0$, and obtain from the above equation that

$$\|\alpha^{(l)}(\mathbf{x})\|^2 = \|\alpha^{(l-1)}(\mathbf{x})\|^2. \tag{25}$$

Apply Eq.(25) back to Eq.(24), we also get

$$\cos\theta^{(l)}(\mathbf{x}, \mathbf{z}) = \frac{\langle \alpha^{(l)}(\mathbf{z}), \alpha^{(l)}(\mathbf{x}) \rangle}{\|\alpha^{(l)}(\mathbf{x})\|\|\alpha^{(l)}(\mathbf{z})\|} = \frac{1}{\pi}\left((\pi - \theta^{(l-1)}(\mathbf{x}, \mathbf{z}))\cos\theta^{(l-1)}(\mathbf{x}, \mathbf{z}) + \sin\theta^{(l-1)}(\mathbf{x}, \mathbf{z})\right)$$
$$\tag{26}$$

That is $\theta^{(l)}(\mathbf{x}, \mathbf{z}) = g(\theta^{(l-1)}(\mathbf{x}, \mathbf{z}))$. Recursively apply this relation, we obtain the desired result. $\quad\square$

## D.3  PROOF OF THEOREM 3.2

*Proof.* By Lemma 3.1, we have that

$$\cos\theta^{(l)}(\mathbf{x}, \mathbf{z}) = \left(1 - \frac{\theta^{(l-1)}(\mathbf{x}, \mathbf{z})}{\pi}\right)\cos\theta^{(l-1)}(\mathbf{x}, \mathbf{z}) + \frac{1}{\pi}\sin\theta^{(l-1)}(\mathbf{x}, \mathbf{z})$$
$$= \cos\theta^{(l-1)}(\mathbf{x}, \mathbf{z})\left(1 + \frac{1}{\pi}\left(\tan\theta^{(l-1)}(\mathbf{x}, \mathbf{z}) - \theta^{(l-1)}(\mathbf{x}, \mathbf{z})\right)\right)$$
$$= \cos\theta^{(l-1)}(\mathbf{x}, \mathbf{z})\left(1 + \frac{1}{3\pi}(\theta^{(l-1)}(\mathbf{x}, \mathbf{z}))^3 + o\left((\theta^{(l-1)}(\mathbf{x}, \mathbf{z}))^3\right)\right).$$

Noting that the Taylor expansion of the $\cos$ function at zero is $\cos z = 1 - \frac{1}{2}z^2 + o(z^3)$, one can easily check that, for all $l \in [L]$,

$$\theta^{(l)}(\mathbf{x}, \mathbf{z}) = \theta^{(l-1)}(\mathbf{x}, \mathbf{z}) - \frac{1}{3\pi}(\theta^{(l-1)}(\mathbf{x}, \mathbf{z}))^2 + o\left((\theta^{(l-1)}(\mathbf{x}, \mathbf{z}))^2\right). \tag{27}$$

Note that $\theta^{(l)}(\mathbf{x}, \mathbf{z}) \leq \theta^{(l-1)}(\mathbf{x}, \mathbf{z}) = o(1/L)$. Iteratively apply the above equation, one gets, for all $l \in [L]$, if $\theta^{(0)}(\mathbf{x}, \mathbf{z}) = o(1/L)$,

$$\theta^{(l)}(\mathbf{x}, \mathbf{z}) = \theta^{(0)}(\mathbf{x}, \mathbf{z}) - \frac{l}{3\pi}(\theta^{(0)}(\mathbf{x}, \mathbf{z}))^2 + o\left((\theta^{(0)}(\mathbf{x}, \mathbf{z}))^2\right). \tag{28}$$

$\square$

## D.4  PROOF OF LEMMA 3.3

*Proof.* The model gradient $\nabla f(\mathbf{x})$ is composed of the components $\nabla_l f(\mathbf{x}) \triangleq \frac{\partial f}{\partial W^l}$, for $l \in [L+1]$. Each such component has the following expression: for $l \in [L+1]$

$$\nabla_l f(\mathbf{x}) = \alpha^{(l-1)}(\mathbf{x})\delta^{(l)}(\mathbf{x}), \tag{29}$$

where

$$\delta^{(l)}(\mathbf{x}) = \left(\frac{2}{m}\right)^{\frac{L-l+1}{2}} W^{(L+1)} \mathbb{I}_{\{\tilde{\alpha}^{(L)}(\mathbf{x}) \geq 0\}} W^{(L)} \mathbb{I}_{\{\tilde{\alpha}^{(L-1)}(\mathbf{x}) \geq 0\}} \cdots W^{(l+1)} \mathbb{I}_{\{\tilde{\alpha}^{(l)}(\mathbf{x}) \geq 0\}}. \quad (30)$$

Note that in Eq.(29), $\nabla_l f(\mathbf{x})$ is an outer product of a column vector $\alpha^{(l-1)}(\mathbf{x}) \in \mathbb{R}^{m_{l-1} \times 1}$ ($m_{l-1} = d$ if $l = 1$, and $m_{l-1} = m$ otherwise) and a row vector $\delta^{(l)}(\mathbf{x}) \in \mathbb{R}^{1 \times m_l}$ ($m_l = 1$ if $l = L+1$, and $m_l = m$ otherwise).

First, we consider the inner product $\langle \nabla_l f(\mathbf{z}), \nabla_l f(\mathbf{x}) \rangle$, for $l \in [L+1]$.[1] By Eq.(29), we have

$$\langle \nabla_l f(\mathbf{z}), \nabla_l f(\mathbf{x}) \rangle = \langle \delta^{(l)}(\mathbf{z}), \delta^{(l)}(\mathbf{x}) \rangle \cdot \langle \alpha^{(l-1)}(\mathbf{z}), \alpha^{(l-1)}(\mathbf{x}) \rangle. \quad (31)$$

For $\langle \alpha^{(l-1)}(\mathbf{z}), \alpha^{(l-1)}(\mathbf{x}) \rangle$, applying Lemma 3.1, we have

$$\langle \alpha^{(l-1)}(\mathbf{z}), \alpha^{(l-1)}(\mathbf{x}) \rangle = \|\mathbf{x}\| \|\mathbf{z}\| \cos \theta^{(l-1)}(\mathbf{x}, \mathbf{z}). \quad (32)$$

For $\langle \delta^{(l)}(\mathbf{z}), \delta^{(l)}(\mathbf{x}) \rangle$, by definition Eq.(30), we have

$$\langle \delta^{(l)}(\mathbf{z}), \delta^{(l)}(\mathbf{x}) \rangle = \left(\frac{2}{m}\right)^{L-l+1}$$
$$\times W^{(L+1)} \mathbb{I}_{\{\tilde{\alpha}^{(L)}(\mathbf{x}) \geq 0\}} \cdots \underbrace{W^{(l+1)} \mathbb{I}_{\{\tilde{\alpha}^{(l)}(\mathbf{x}) \geq 0, \tilde{\alpha}^{(l)}(\mathbf{z}) \geq 0\}} W^{(l+1)T}}_{A} \cdots \mathbb{I}_{\{\tilde{\alpha}^{(L)}(\mathbf{z}) \geq 0\}} W^{(L+1)T}$$

Recalling that $\tilde{\alpha}^{(l)} = W^{(l)} \tilde{\alpha}^{(l-1)}$ and applying Lemma D.3 on the the term $A$ above, we obtain

$$\langle \delta^{(l)}(\mathbf{z}), \delta^{(l)}(\mathbf{x}) \rangle = \frac{\pi - \theta^{(l-1)}(\mathbf{x}, \mathbf{z})}{\pi} \langle \delta^{(l+1)}(\mathbf{z}), \delta^{(l+1)}(\mathbf{x}) \rangle.$$

Recursively applying the above formula for $l' = l, l+1, \cdots, L$, and noticing that $\delta^{(L+1)} = 1$, we have

$$\langle \delta^{(l)}(\mathbf{z}), \delta^{(l)}(\mathbf{x}) \rangle = \prod_{l'=l-1}^{L+1} \left(1 - \frac{\theta^{(l')}(\mathbf{x}, \mathbf{z})}{\pi}\right). \quad (33)$$

Combining Eq.(31), (32) and (33), we have

$$\langle \nabla_l f(\mathbf{z}), \nabla_l f(\mathbf{x}) \rangle = \|\mathbf{x}\| \|\mathbf{z}\| \cos \theta^{(l-1)}(\mathbf{x}, \mathbf{z}) \prod_{l'=l-1}^{L-1} \left(1 - \frac{\theta^{(l')}(\mathbf{x}, \mathbf{z})}{\pi}\right). \quad (34)$$

For the inner product between the full model gradients, we have

$$\langle \nabla f(\mathbf{z}), \nabla f(\mathbf{x}) \rangle = \sum_{l=1}^{L+1} \langle \nabla_l f(\mathbf{z}), \nabla_l f(\mathbf{x}) \rangle = \|\mathbf{x}\| \|\mathbf{z}\| \sum_{l=0}^{L} \left[\cos \theta^{(l)}(\mathbf{x}, \mathbf{z}) \prod_{l'=l}^{L-1} \left(1 - \frac{\theta^{(l')}(\mathbf{x}, \mathbf{z})}{\pi}\right)\right]. \quad (35)$$

Putting $\mathbf{x} = \mathbf{z}$ in the above equation, we have $\theta^{(l)}(\mathbf{x}, \mathbf{z}) = 0$ for all $l \in [L]$, and obtain

$$\|\nabla f(\mathbf{x})\|^2 = \|\mathbf{x}\|^2 \cdot (L+1). \quad (36)$$

Hence, we have

$$\cos \phi(\mathbf{x}, \mathbf{z}) = \frac{\langle \nabla f(\mathbf{z}), \nabla f(\mathbf{x}) \rangle}{\|\nabla f(\mathbf{x})\| \|\nabla f(\mathbf{z})\|} = \frac{1}{L+1} \sum_{l=0}^{L} \left[\cos \theta^{(l)}(\mathbf{x}, \mathbf{z}) \prod_{l'=l}^{L-1} (1 - \theta^{(l')}(\mathbf{x}, \mathbf{z})/\pi)\right]. \quad (37)$$

$\square$

---

[1] With a bit of abuse of notation, we refer to the flattened vectors of $\nabla_l f$ in the inner product.

### D.5 PROOF OF THEOREM 3.4

*Proof.* For simplicity of notation, we don't explicitly write out the dependent on the inputs $\mathbf{x}, \mathbf{z}$, and write $\theta^{(l)} \triangleq \theta^{(l)}(\mathbf{x}, \mathbf{z})$, and $\phi \triangleq \phi(\mathbf{x}, \mathbf{z})$. We start the proof with the relation provided by Lemma 3.3.

$$
\begin{aligned}
\cos\phi(\mathbf{x}, \mathbf{z}) &= \frac{1}{L+1} \sum_{l=0}^{L} \left[ \cos\theta^{(l)} \prod_{l'=l}^{L-1} (1 - \theta^{(l')}/\pi) \right] \\
&\overset{(a)}{=} \frac{1}{L+1} \sum_{l=0}^{L} \left[ \cos\theta^{(0)} \prod_{l'=0}^{l-1} \left( 1 + \frac{1}{\pi}\tan\theta^{(l')} - \frac{1}{\pi}\theta^{(l')} \right) \prod_{l'=l}^{L-1} (1 - \theta^{(l')}/\pi) \right] \\
&\overset{(b)}{=} \frac{1}{L+1} \sum_{l=0}^{L} \left[ \cos\theta^{(0)} \prod_{l'=0}^{l-1} \left( 1 + \frac{1}{3\pi}(\theta^{(l')})^3 + o(\theta^{(l')})^3 \right) \prod_{l'=l}^{L-1} (1 - \theta^{(l')}/\pi) \right] \\
&\overset{(c)}{=} \frac{\cos\theta^{(0)}}{L+1} \sum_{l=0}^{L} \left[ \prod_{l'=0}^{l-1} \left( 1 + \frac{1}{3\pi}(\theta^{(0)})^3 + o(\theta^{(0)})^3 \right) \right. \\
&\qquad\qquad \left. \times \prod_{l'=l}^{L-1} \left( 1 - \frac{1}{\pi}\theta^{(0)} + \frac{l'}{3\pi^2}(\theta^{(0)})^2 + o((\theta^{(0)})^2) \right) \right] \\
&= \frac{\cos\theta^{(0)}}{L+1} \sum_{l=0}^{L} \left( 1 - \frac{L-l}{\pi}\theta^{(0)} + \frac{(L-l)(2L-l-2)}{3\pi^2}(\theta^{(0)})^2 + o((\theta^{(0)})^2) \right) \\
&= \cos\theta^{(0)} \left( 1 - \frac{L}{2}\theta^{(0)} + o(\theta^{(0)}) \right).
\end{aligned}
$$

$\square$

### D.6 PROOF OF THEOREM 4.2

*Proof.* First of all, we note that in scenario (b), i.e., the network with all ReLU activation removed, the network simply becomes a linear neural network (while with the same trainable parameters $W^{(1)}$ as the ReLU network in scenario (a)). By the analysis in Section 2.1, we can easily see that the NTK matrix in scenario (b) is equivalent to the Gram matrix $G$, and $\kappa_b = \kappa_0$. Hence, whenever comparing the two scenarios, it suffices to compare the NTK $K$ (and its condition number $\kappa$) of ReLU network with the Gram matrix $G$ (and its condition number $\kappa_0$).

We prove the theorem by induction.

**Base case: ReLU neural network of depth $L = 1$.** First, consider the shallow ReLU neural network

$$
f(W; \mathbf{x}) = \frac{\sqrt{2}}{\sqrt{m}} \mathbf{v}^T \sigma(W\mathbf{x}), \tag{38}
$$

where $W$ are the trainable parameters.

The model gradient, for an arbitrary input $\mathbf{x}$, can be written as

$$
\nabla f(\mathbf{x}) = \mathbf{x}\delta(\mathbf{x}) \in \mathbb{R}^{d \times m}, \tag{39}
$$

where $\delta(\mathbf{x}) \in \mathbb{R}^{1 \times m}$ has the following expression

$$
\delta(\mathbf{x}) = \sqrt{\frac{2}{m}} \mathbf{v}^T \mathbb{I}_{\{W\mathbf{x} \geq 0\}}.
$$

At initialization, $W$ is a random matrix. Recall that the NTK $K = FF^T$, where the gradient feature matrix $F$ consist of the gradient feature vectors $\nabla f(\mathbf{x})$ for all $\mathbf{x}$ for the dataset. Applying Lemma C.1 in the limit of $m \to \infty$, we have that each entry $K_{ij}$ is equivalent to $\sum_{k,l} \mathbf{A}'_{ikl}\mathbf{A}'_{jkl}$, with $\mathbf{A}'_{ikl} := \sqrt{2}X_{ik}\mathbb{I}_{\{W_{l:}X_{i:} \geq 0\}}$, where $X \in \mathbb{R}^{n \times d}$ is the matrix of input data. Then apply Lemma D.4, we immediately have that

$$
\lambda_{min}(K) > \lambda_{min}(G), \quad \lambda_{max}(K) < \lambda_{max}(G).
$$

Hence, we have that $\kappa_a < \kappa_b$.

In addition, note that this network has one hidden layer, and that the "zero-hidden layer" network is just simply the linear model. For linear model, the NTK is simply the Gram matrix. Hence, for the base case, we have $\kappa_{f_1} < \kappa_{f_2} = \kappa_0$, with network $f_1$ of depth 1 and network $f_2$ of depth 0.

**Induction hypothesis.** Suppose that, for a ReLU network $f_{L-1}$ of depth $L-1$, its NTK condition number $\kappa_{L-1}$ is strictly smaller than $\kappa_0$.

**Induction step.** Now, let's consider the two ReLU networks $f_L$ of depth $L$ and $f_{L-1}$. It is suffices to prove that $\kappa_L < \kappa_{L-1}$. The model gradients, for any given input $\mathbf{x}$, can be written as:

$$\nabla f_L(\mathbf{x}) = \mathbf{x}\delta_L(\mathbf{x}) \in \mathbb{R}^{d\times m}, \quad \nabla f_{L-1}(\mathbf{x}) = \mathbf{x}\delta_{L-1}(\mathbf{x}) \in \mathbb{R}^{d\times m},$$

where

$$\delta_L(\mathbf{x}) = \sqrt{\frac{2}{m}}W^{(L+1)}\mathbb{I}_{\{W^{(L)}\alpha^{(L-1)}\geq 0\}}\sqrt{\frac{2}{m}}W^{(L)}\mathbb{I}_{\{W^{(L-1)}\alpha^{(L-2)}\geq 0\}}\cdots\sqrt{\frac{2}{m}}W^{(2)}\mathbb{I}_{\{W^{(1)}\alpha^{(0)}\geq 0\}}$$

$$\delta_{L-1}(\mathbf{x}) = \sqrt{\frac{2}{m}}W^{(L)}\mathbb{I}_{\{W^{(L-1)}\alpha^{(L-2)}\geq 0\}}\cdots\sqrt{\frac{2}{m}}W^{(2)}\mathbb{I}_{\{W^{(1)}\alpha^{(0)}\geq 0\}}$$

Note that the matrix $W^{(L)}$ has different dimensions for $f_L$ and $f_{L-1}$.

Using the same argument as in the base case, as well as applying Lemma C.1 when contracting the $\delta(\mathbf{x})$'s, we directly obtain $\kappa_L < \kappa_{L-1}$. $\qquad\square$

## D.7 PROOF OF THEOREM 4.3

*Proof.* First, let's consider the scenario (a), i.e. the ReLU network with ReLU *unremoved*. According to the definition of NTK and Lemma 3.3, the NTK matrix $K$ for this dataset $\mathcal{D} = \{(\mathbf{x}_1, y_1), (\mathbf{x}_2, y_2)\}$ is (NTK is normalized by the factor $1/(L+1)^2$):

$$K = \begin{pmatrix} \|\nabla f(\mathbf{x}_1)\|^2 & \langle\nabla f(\mathbf{x}_1), \nabla f(\mathbf{x}_2)\rangle \\ \langle\nabla f(\mathbf{x}_2), \nabla f(\mathbf{x}_1)\rangle & \|\nabla f(\mathbf{x}_2)\|^2 \end{pmatrix} = \begin{pmatrix} \|\mathbf{x}_1\|^2 & \|\mathbf{x}_1\|\|\mathbf{x}_2\|\cos\phi \\ \|\mathbf{x}_1\|\|\mathbf{x}_2\|\cos\phi & \|\mathbf{x}_2\|^2 \end{pmatrix}.$$

The eigenvalues of the NTK matrix $K$ are given by

$$\lambda_1(K) = \frac{1}{2}\left(\|\mathbf{x}_1\|^2 + \|\mathbf{x}_2\|^2 + \sqrt{\|\mathbf{x}_1\|^4 + \|\mathbf{x}_2\|^4 + \|\mathbf{x}_1\|^2\|\mathbf{x}_2\|^2\cos 2\phi}\right), \tag{40a}$$

$$\lambda_2(K) = \frac{1}{2}\left(\|\mathbf{x}_1\|^2 + \|\mathbf{x}_2\|^2 - \sqrt{\|\mathbf{x}_1\|^4 + \|\mathbf{x}_2\|^4 + \|\mathbf{x}_1\|^2\|\mathbf{x}_2\|^2\cos 2\phi}\right). \tag{40b}$$

In the scenario (b), the ReLU activation is removed in the network, resulting in a linear neural network. In this case, the NTK is equivalent to the Gram matrix $G$, as given by Corollary 2.3. We have

$$G = \begin{pmatrix} \|\mathbf{x}_1\|^2 & \mathbf{x}_1^T\mathbf{x}_2 \\ \mathbf{x}_1^T\mathbf{x}_2 & \|\mathbf{x}_2\|^2 \end{pmatrix} = \begin{pmatrix} \|\mathbf{x}_1\|^2 & \|\mathbf{x}_1\|\|\mathbf{x}_2\|\cos\theta_{in} \\ \|\mathbf{x}_1\|\|\mathbf{x}_2\|\cos\theta_{in} & \|\mathbf{x}_2\|^2 \end{pmatrix},$$

and its eigenvalues as

$$\lambda_1(G) = \frac{1}{2}\left(\|\mathbf{x}_1\|^2 + \|\mathbf{x}_2\|^2 + \sqrt{\|\mathbf{x}_1\|^4 + \|\mathbf{x}_2\|^4 + \|\mathbf{x}_1\|^2\|\mathbf{x}_2\|^2\cos 2\theta_{in}}\right),$$

$$\lambda_2(G) = \frac{1}{2}\left(\|\mathbf{x}_1\|^2 + \|\mathbf{x}_2\|^2 - \sqrt{\|\mathbf{x}_1\|^4 + \|\mathbf{x}_2\|^4 + \|\mathbf{x}_1\|^2\|\mathbf{x}_2\|^2\cos 2\theta_{in}}\right).$$

By Theorem 3.4, we have $\cos\phi < \cos\theta_{in}$, when $\theta_{in} = o(1)$ and $\theta_{in} \neq 0$. Hence, we have the following relations

$$\lambda_1(G) > \lambda_1(K) > \lambda_2(K) > \lambda_2(G),$$

which immediately implies $\kappa_a < \kappa_b$.

When comparing ReLU networks with different depths, i.e., network $f_1$ with depth $L_1$ and network $f_2$ with depth $L_2$ with $L_1 > L_2$, notice that in Eq.(40) the top eigenvalue $\lambda_1$ monotonically decreases in $\phi$, and the bottom (smaller) eigenvalue $\lambda_2$ monotonically increases in $\phi$. By Theorem 3.4, we

know that the deeper ReLU network $f_1$ has a better data separation than the shallower one $f_2$, i.e., $\phi_{f_1} > \phi_{f_2}$. Hence, we get

$$\lambda_1(K_{f_2}) > \lambda_1(K_{f_1}) > \lambda_2(K_{f_1}) > \lambda_2(K_{f_2}). \tag{41}$$

Therefore, we obtain $\kappa_{f_1} < \kappa_{f_2}$. Namely the deeper ReLU network has a smaller NTK condition number. $\qquad\square$

## E  TECHNICAL PROOFS

### E.1  PROOF OF LEMMA C.1

*Proof.* We denote $A_{ij}$ as the $(i,j)$-th entry of the matrix $A$. Therefore, $(A^T A)_{ij} = \sum_{k=1}^m A_{ki} A_{kj}$. First we find the mean of each $(A^T A)_{ij}$. Since $A_{ij}$ are i.i.d. and has zero mean, we can easily see that for any index $k$,

$$\mathbb{E}[A_{ki} A_{kj}] = \begin{cases} 1, & \text{if } i = j \\ 0, & \text{otherwise} \end{cases}.$$

Consequently,

$$\mathbb{E}[(\frac{1}{m} A^T A)_{ij}] = \begin{cases} 1, & \text{if } i = j \\ 0, & \text{otherwise} \end{cases}.$$

That is $\mathbb{E}[\frac{1}{m} A^T A] = I_d$.

Now we consider the variance of each $(A^T A)_{ij}$. If $i \neq j$ we can explicitly write,

$$\begin{aligned}
Var\left[\frac{1}{m}(A^T A)_{ij}\right] &= \frac{1}{m^2} \cdot \mathbb{E}\left[\sum_{k_1=1}^m \sum_{k_2=1}^m A_{k_1 i} A_{k_1 j} A_{k_2 i} A_{k_2 j}\right] \\
&= \frac{1}{m^2} \cdot \sum_{k_1=1}^m \sum_{k_2=1}^m \mathbb{E}\left[A_{k_1 i} A_{k_1 j} A_{k_2 i} A_{k_2 j}\right] \\
&= \frac{1}{m^2}\left(\sum_{k=1}^m \mathbb{E}\left[A_{ki}^2 A_{kj}^2\right] + \sum_{k_1 \neq k_2} \mathbb{E}\left[A_{k_1 i} A_{k_1 j} A_{k_2 i} A_{k_2 j}\right]\right) \\
&= \frac{1}{m^2}\left(\sum_{k=1}^m \mathbb{E}\left[A_{ki}^2\right] \mathbb{E}\left[A_{kj}^2\right] + \sum_{k_1 \neq k_2} \mathbb{E}[A_{k_1 i}] \mathbb{E}[A_{k_1 j}] \mathbb{E}[A_{k_2 i}] \mathbb{E}[A_{k_2 j}]\right) \\
&= \frac{1}{m^2} \cdot (m + 0) = \frac{1}{m}.
\end{aligned}$$

In the case of $i = j$, then,

$$Var\left[\frac{1}{m}(A^T A)_{ii}\right] = \frac{1}{m^2} \cdot Var\left[\sum_{k=1}^m A_{ki}^2\right] = \frac{1}{m^2} \cdot \sum_{k=1}^m Var\left[A_{ki}^2\right] \overset{(a)}{=} \frac{1}{m^2}(m \cdot 2) = \frac{2}{m}. \tag{42}$$

In the equality (a) above, we used the fact that $A_{ki}^2 \sim \chi^2(1)$. Therefore, $\lim_{m \to \infty} Var(\frac{1}{m}(A^T A)) = 0$.

Now applying Chebyshev's inequality we get,

$$Pr(|\frac{1}{m} A^T A - I_d| \geq \epsilon) \leq \frac{Var(\frac{1}{m}(A^T A))}{\epsilon} \tag{43}$$

Obviously for any $\epsilon \geq 0$ as $m \to \infty$, the R.H.S. goes to zero. Thus, $\frac{1}{m} A^T A \to I_{d \times d}$, in probability. $\qquad\square$

## E.2 PROOF OF LEMMA D.1

*Proof.* Note that the random vector $\mathbf{w}$ is isotropically distributed and that only inner products $\mathbf{w}^T\mathbf{v}_1$ and $\mathbf{w}^T\mathbf{v}_2$ appear, hence we can assume without loss of generality that (if not, one can rotate the coordinate system to make it true):

$$\mathbf{v}_1 = \|\mathbf{v}_1\|(1, 0, 0, \cdots, 0),$$
$$\mathbf{v}_2 = \|\mathbf{v}_2\|(\cos\theta, \sin\theta, 0, \cdots, 0).$$

In this setting, the only relevant parts of $\mathbf{w}$ are its first two scalar components $w_1$ and $w_2$. Define $\tilde{\mathbf{w}}$ as

$$\tilde{\mathbf{w}} = (w_1, w_2, 0, \cdots, 0) = \sqrt{w_1^2 + w_2^2}(\cos\omega, \sin\omega, 0, \cdots, 0). \tag{44}$$

Then,

$$\mathbb{P}[(\mathbf{w}^T\mathbf{v}_1 \geq 0) \wedge (\mathbf{w}^T\mathbf{v}_2 \geq 0)] = \mathbb{P}[(\tilde{\mathbf{w}}^T\mathbf{v}_1 \geq 0) \wedge (\tilde{\mathbf{w}}^T\mathbf{v}_2 \geq 0)] = \frac{1}{2\pi}\int_{\theta-\frac{\pi}{2}}^{\frac{\pi}{2}} d\omega = \frac{1}{2} - \frac{\theta}{2\pi}.$$

$\square$

## E.3 PROOF OF LEMMA D.2

*Proof.* Note that the ReLU activation function $\sigma(z)$ can be written as $z\mathbb{I}_{z\geq 0}$. We have,

$$\langle \mathbf{u}_1, \mathbf{u}_2 \rangle = \frac{2}{q}\mathbf{v}_1^T W^T \mathbb{I}_{\{W\mathbf{v}_1 \geq 0, W\mathbf{v}_2 \geq 0\}} W\mathbf{v}_2$$

$$= \frac{2}{q}\sum_{i=1}^{q} \mathbf{v}_1^T (W_{i\cdot})^T \mathbb{I}_{\{W_{i\cdot}\mathbf{v}_1 \geq 0, W_{i\cdot}\mathbf{v}_2 \geq 0\}} W_{i\cdot}\mathbf{v}_2$$

$$\overset{q\to\infty}{=} 2\mathbb{E}_{\mathbf{w}\sim\mathcal{N}(0,I_{p\times p})}[\mathbf{v}_1^T\mathbf{w}\mathbb{I}_{\{\mathbf{w}^T\mathbf{v}_1 \geq 0, \mathbf{w}^T\mathbf{v}_2 \geq 0\}}\mathbf{w}^T\mathbf{v}_2]$$

Note that the random vector $\mathbf{w}$ is isotropically distributed and that only inner products $\mathbf{w}^T\mathbf{v}_1$ and $\mathbf{w}^T\mathbf{v}_2$ appear, hence we can assume without loss of generality that (if not, one can rotate the coordinate system to make it true):

$$\mathbf{v}_1 = \|\mathbf{v}_1\|(1, 0, 0, \cdots, 0),$$
$$\mathbf{v}_2 = \|\mathbf{v}_2\|(\cos\theta, \sin\theta, 0, \cdots, 0).$$

In this setting, the only relevant parts of $\mathbf{w}$ are its first two scalar components $w_1$ and $w_2$. Define $\tilde{\mathbf{w}}$ as

$$\tilde{\mathbf{w}} = (w_1, w_2, 0, \cdots, 0) = \sqrt{w_1^2 + w_2^2}(\cos\omega, \sin\omega, 0, \cdots, 0). \tag{45}$$

Then, in the limit of $q \to \infty$,

$$\langle \mathbf{u}_1, \mathbf{u}_2 \rangle = 2\mathbb{E}_{\mathbf{w}\sim\mathcal{N}(0,I_{p\times p})}[\mathbf{v}_1^T\mathbf{w}\mathbb{I}_{\{\mathbf{w}^T\mathbf{v}_1 \geq 0, \mathbf{w}^T\mathbf{v}_2 \geq 0\}}\mathbf{w}^T\mathbf{v}_2]$$

$$= 2\mathbb{E}_{\tilde{\mathbf{w}}\sim\mathcal{N}(0,I_{2\times 2})}[\mathbf{v}_1^T\tilde{\mathbf{w}}\mathbb{I}_{\{\tilde{\mathbf{w}}^T\mathbf{v}_1 \geq 0, \tilde{\mathbf{w}}^T\mathbf{v}_2 \geq 0\}}\tilde{\mathbf{w}}^T\mathbf{v}_2]$$

$$= 2\|\mathbf{v}_1\|\|\mathbf{v}_2\| \cdot \mathbb{E}_{\tilde{\mathbf{w}}\sim\mathcal{N}(0,I_{2\times 2})}[\|\tilde{\mathbf{w}}\|^2] \cdot \frac{1}{2\pi}\int_{\theta-\frac{\pi}{2}}^{\frac{\pi}{2}} \cos\omega\cos(\theta-\omega)d\omega$$

$$= 2\|\mathbf{v}_1\|\|\mathbf{v}_2\| \cdot 2 \cdot \frac{1}{4\pi}\left((\pi-\theta)\cos\theta + \sin\theta\right)$$

$$= \|\mathbf{v}_1\|\|\mathbf{v}_2\|\frac{1}{\pi}\left((\pi-\theta)\cos\theta + \sin\theta\right).$$

$\square$

### E.4 PROOF OF LEMMA D.3

*Proof.*

$$A_1 A_2^T = \frac{2}{q} \sum_{k=1}^{q} U_{\cdot k} \mathbb{I}_{\{W_{k\cdot} \mathbf{v}_1 \geq 0, W_{k\cdot} \mathbf{v}_2 \geq 0\}} (U_{\cdot k})^T$$

$$\overset{q \to \infty}{=} 2 \cdot \mathbb{E}_{\mathbf{u} \sim \mathcal{N}(0, I_{s \times s}), \mathbf{w} \sim \mathcal{N}(0, I_{p \times p})} [\mathbf{u}\mathbf{u}^T \mathbb{I}_{\{\mathbf{w}^T \mathbf{v}_1 \geq 0, \mathbf{w}^T \mathbf{v}_2 \geq 0\}}]$$

$$\overset{(a)}{=} 2 \cdot \mathbb{E}_{\mathbf{u} \sim \mathcal{N}(0, I_{s \times s})} [\mathbf{u}\mathbf{u}^T] \cdot \mathbb{E}_{\mathbf{w} \sim \mathcal{N}(0, I_{p \times p})} [\mathbb{I}_{\{\mathbf{w}^T \mathbf{v}_1 \geq 0, \mathbf{w}^T \mathbf{v}_2 \geq 0\}}]$$

$$= 2 \cdot \mathbb{E}_{\mathbf{u} \sim \mathcal{N}(0, I_{s \times s})} [\mathbf{u}\mathbf{u}^T] \cdot \mathbb{P}[(\mathbf{w}^T \mathbf{v}_1 \geq 0) \wedge (\mathbf{w}^T \mathbf{v}_2 \geq 0)]$$

$$\overset{(b)}{=} \frac{\pi - \theta}{\pi} I_{s \times s}.$$

In the step $(a)$ above, we used the fact that $U$ is independent of $W$, $\mathbf{v}_1$ and $\mathbf{v}_2$. In the step $(b)$ above, we applied Lemma D.1, and used the fact that $\mathbb{E}_{\mathbf{u} \sim \mathcal{N}(0, I_{s \times s})} [\mathbf{u}\mathbf{u}^T] = I_{s \times s}$. $\qquad \square$

### E.5 PROOF OF LEMMA D.4

*Proof.* Starting from the definition of the smallest eigenvalue, we have that $\lambda_{min}(B')$ satisfies

$$\lambda_{min}(B') = \min_{\mathbf{u} \neq 0} \frac{\mathbf{u}^T B' \mathbf{u}}{\|\mathbf{u}\|^2}$$

$$= \min_{\mathbf{u} \neq 0} \frac{\sum_{l=1}^{q} \sum_{k=1}^{p} (\sum_{i=1}^{n} \sqrt{2} u_i A_{ik} \mathbb{I}_{\{W_{l:} A_{i:} \geq 0\}})^2}{\sum_{i=1}^{n} u_i^2}$$

$$= \min_{\mathbf{u} \neq 0} \sum_{l=1}^{q} \frac{\sum_{i=1}^{n} 2(u_i \mathbb{I}_{\{W_{l:} A_{i:} \geq 0\}})^2}{\sum_{i=1}^{n} u_i^2} \frac{\sum_{k=1}^{p} (\sum_{i=1}^{n} \sqrt{2} u_i A_{ik} \mathbb{I}_{\{W_{l:} A_{i:} \geq 0\}})^2}{\sum_{i=1}^{n} 2(u_i \mathbb{I}_{\{W_{l:} A_{i:} \geq 0\}})^2}$$

$$\overset{(a)}{>} \min_{\mathbf{u} \neq 0} \sum_{l=1}^{q} \frac{\sum_{i=1}^{n} 2(u_i \mathbb{I}_{\{W_{l:} A_{i:} \geq 0\}})^2}{\sum_{i=1}^{n} u_i^2} \lambda_{min}(B). \tag{46}$$

In the inequality $(a)$ above, we made the following treatment: for each fixed $l$, we consider $u_i \mathbb{I}_{\{W_{l:} A_{i:} \geq 0\}}$ as the $i$-th component of a vector $\mathbf{u}'_l$; by definition, the minimum eigenvalue of matrix $B = AA^T$

$$\lambda_{min}(B) = \min_{\mathbf{u}' \neq 0} (\mathbf{u}')^T B \mathbf{u}' / \|\mathbf{u}'\|^2 \leq (\mathbf{u}'_j)^T B \mathbf{u}'_j / \|\mathbf{u}'_j\|^2, \quad \forall j; \tag{47}$$

moreover, this $\leq$ inequality becomes equality, if and only if all $\mathbf{u}'_j$ are the same and equal to $\arg\min_{\mathbf{u}' \neq 0} (\mathbf{u}')^T G \mathbf{u}' / \|\mathbf{u}'\|^2$. It is easy to see, when the dataset is not degenerate, for different $j$, $\mathbf{u}'_j$ are different, hence only the strict inequality $<$ holds in step $(a)$.

Continuing from Eq.(46), we have

$$\lambda_{min}(B') > \min_{\mathbf{u} \neq 0} \sum_{l=1}^{q} \frac{\sum_{i=1}^{n} 2(u_i \mathbb{I}_{\{\{W_{l:} A_{i:} \geq 0\}})^2}{\sum_{i=1}^{n} u_i^2} \lambda_{min}(B)$$

$$= \min_{\mathbf{u} \neq 0} \frac{\sum_{i=1}^{n} 2 u_i^2 \sum_{l=1}^{q} \mathbb{I}_{\{\{W_{l:} A_{i:} \geq 0\})}}{\sum_{i=1}^{n} u_i^2} \lambda_{min}(B)$$

$$= \min_{\mathbf{u} \neq 0} \frac{\sum_{i=1}^{n} u_i^2}{\sum_{i=1}^{n} u_i^2} \lambda_{min}(B) = \lambda_{min}(B).$$

Therefore, we have that $\lambda_{min}(B') > \lambda_{min}(B)$.

As for the largest eigenvalue $\lambda_{max}(B')$, we can apply the same logic above for $\lambda_{min}(K)$ (except replacing the $\min$ operator by $\max$ and have $<$ in step $(a)$) to get $\lambda_{max}(B') < \lambda_{max}(B)$. $\qquad \square$

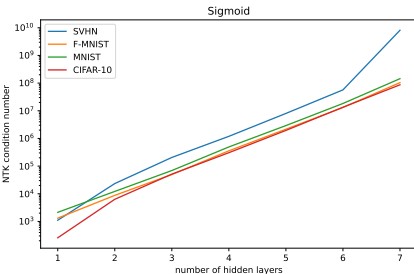 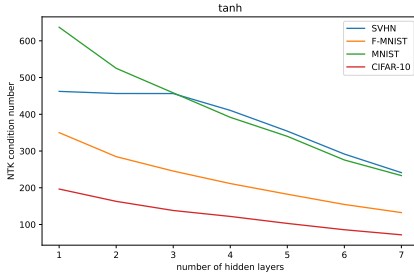

Figure 5: NTK condition number vs. depth, for *sigmoid*-activated network and *tanh*-activated network.

## F    NUMERICAL RESULTS OF OTHER ACTIVATION FUNCTIONS

In this section, we show some preliminary numerical results of some other non-linear activation functions, although the main focus of this paper is ReLU.

Specifically, analogous to what we did for ReLU network, we compute the NTK condition number for the following two types of non-linearly activated neural networks at random initialization: *sigmoid*-activated network and *tanh*-activated network. In both cases, we use the same network width, 512, as in Figure 2 for ReLU network. The scaling factor, $\sqrt{2/m_l}$ in Eq.(1), was replaced by $\sqrt{c_\sigma/m_l}$, where $c_\sigma$ is a activation-specific constant and is defined as $c_\sigma = \left(\mathbb{E}_{x\sim\mathcal{N}(0,1)}[\sigma(x)^2]\right)^{-1}$ (see for example Eq.(2) of Du et al. (2019)).

Figure 5 shows the dependence of the NTK condition number on the network depth. We observe that different non-linear activation function may have different effects on the NTK condition numbers $\kappa$. As the figure tells, *tanh* also helps to decrease the condition number (similar to ReLU), while *sigmoid* has the opposite effect, worsening the NTK conditioning.

A theoretical analysis of these non-linear activation functions are out of the scope of this paper, but we expect future work will theoretically clarify the exact effects of different types of non-linear activation functions.

