# OpenReview forum: "Non-linear activation soothes NTK conditioning for wide neural networks: a study in the ReLU case"
_ICLR.cc/2025/Conference — Submitted to ICLR 2025_

### Official Review · Reviewer_ZNtx · 2024-10-16

**Soundness:** 3
**Presentation:** 3
**Contribution:** 3
**Rating:** 6
**Confidence:** 2

**Summary:**

This paper theoretically studies the beneficial effects and interesting properties of ReLU activation function.

**Strengths:**

The strength of this paper is to show that ReLU activation function has the effects of better data separation and better NTK condition. This paper implies the optimization benefit that ReLU network helps improving worst case convergence rate of gradient descent and faster convergence than shallower one.

**Weaknesses:**

As mentioned in Conclusion and Discussion, the finite depth case is focused, and not directly extended to the infinite depth case.

**Questions:**

Out of interest, can the analysis of this paper be applied to only ReLU ? In other words, does this paper use specific properties of ReLU in the proof? For example, can it be little bit generalized to Leaky ReLU (= $ax$ when $x<0$, and $x$ when $x \geq 0$. The case when $a=0$ is special case of ReLU) ?

---

### Official Review · Reviewer_RnSE · 2024-10-22

**Soundness:** 3
**Presentation:** 3
**Contribution:** 2
**Rating:** 5
**Confidence:** 4

**Summary:**

This paper investigates the impact of non-linear activation functions, specifically the ReLU in wide neural networks. The authors demonstrate that ReLU activation improves data separation in feature space and enhances the conditioning of the NTK, leading to better theoretical convergence rates of gradient descent optimization algorithms.

**Strengths:**

The paper provides a thorough theoretical analysis backed by empirical evidence demonstrating that ReLU activation improves both the separation of data in feature space and the conditioning of the NTK.

**Weaknesses:**

The analysis is specifically focused on networks with ReLU activation and the results primarily demonstrate that ReLU NTK outperforms linear NTK, which may seem somewhat limited in scope.


Typo: Line 209 $\nabla f(x)(z) \to \nabla f(z)$

**Questions:**

1. Can the findings be generalized to other non-linear activation functions? How might the NTK conditioning change with different functions?

2. What are the implications of these findings on network architecture design? Specifically, how might they influence decisions on depth and width of networks (m is finite)?

---

### Official Review · Reviewer_RSh8 · 2024-11-02

**Soundness:** 3
**Presentation:** 3
**Contribution:** 2
**Rating:** 5
**Confidence:** 4

**Summary:**

This paper compares the deep networks with or without the ReLU activation under the NTK regime. They show that ReLU has two effects: (a) There is a larger angle separation for similar data in the feature space; (b) The NTK conditional better becomes larger.  They also show that the depth of the network will further enhance these effects.

**Strengths:**

- The paper is well-written, and the claims appear to be sound.
- The experiments are comprehensive and align well with the theoretical results.
- The investigation of the angle between two samples after projection into the feature space is both novel and intriguing.

**Weaknesses:**

- This paper compares only ReLU networks and linear networks. The results are not surprising, given the established fact that non-linear activations enhance the expressivity of networks.

- The title mentions "Non-Linear Activation Soothes NTK Condition," but the paper focuses solely on ReLU, which is just one type of non-linear activation.

- The NTK regime typically requires the network width to exceed a certain constant. However, the paper assumes that the width approaches infinity. It would be beneficial if the authors could relax this condition.

**Questions:**

- Could the authors compare additional non-linear activation functions in the experiments?

- Is it feasible to extend the current analysis to GeLU or SiLU?

- Can the condition of infinite width be relaxed to require a sufficiently large width?

- There is a typo in line 195; $G$ should be in $\mathbb{R}^{n \times n}$.
 .

---

### Official Review · Reviewer_zKSn · 2024-11-04

**Soundness:** 3
**Presentation:** 3
**Contribution:** 3
**Rating:** 6
**Confidence:** 3

**Summary:**

In this work, the authors study the benefits of using ReLU activation for Wide Feedforward Neural Networks under the NTK framework. Contrary to previous works that focused on expressivity, they adopt a novel perspective and show that ReLU activation yields better data separation in the gradient feature space and, hence, better NTK conditioning when compared to Linear Networks. This effect is even exacerbated with deeper networks. They also illustrate their main results with experiments on synthetic and benchmark datasets (MNIST, etc.).

**Strengths:**

Approaching the study from the perspective of data separability rather than focusing on expressivity proves to be an insightful choice. The insights obtained are interesting and complement the existing results well. Besides, the paper is well-written and accessible to a relatively broad audience. The experiments illustrate well the main findings.

**Weaknesses:**

The main limitation I observed, which is often anticipated in papers leveraging the NTK framework, is that this initialization differs from those commonly used in practice. While it allows for theoretical insights, the paper would be significantly strengthened if the authors could provide empirical verification to determine if these findings extend to more practical initialization schemes.

A secondary limitation lies in Theorems 4.2 and 4.3, which establish that enhanced data separability in the gradient feature space concretely benefits NTK conditioning. However, these results rest on stronger assumptions, though the experiments partially compensate for this limitation.

**Questions:**

- In the paragraph beginning on line 132, the authors reference a paper by Arora et al., which suggests that deep linear networks accelerate optimization. This claim appears to contradict the message of Section 2 in the paper. A brief comment could clarify this point and help readers better reconcile these perspectives.

- I would suggest expanding the 'Infinite Width Limit' section (line 177) by adding a couple of sentences to clarify what is meant by taking the infinite limit. Specifically, it would be helpful for the authors to specify the type of convergence they refer to and how they manage successive layers in this context. As stated in the theorems ($m \rightarrow +\infty$), it seems to imply that the widths of different layers go to infinity simultaneously. However, after a high-level check of the proofs, it appears some arguments use induction on the layers, taking the limit successively, one layer at a time. Adding clarification here would improve reader comprehension and strengthen the rigor of the presentation.

---

### Meta-Review · Area_Chair_9iyr · 2024-12-19

**Metareview:**

Dear Authors,

Thank you for your valuable contribution to ICLR and the ML community. Your submitted paper has undergone a rigorous review process, and I have carefully read and considered the feedback provided by the reviewers.

This work investigates the impact of non-linearities in (infinitely) wide neural networks, demonstrating that ReLU activation improves data separation in feature space and enhances the conditioning of the Neural Tangent Kernel (NTK).

The paper received borderline final review scores (5,5,6,6). Certain critical issues were raised including (i) limited novelty of the conclusion that ReLU NTK is better than the linear NTK  (ii) limited scope of the framework (e.g., type of activations) (iii) limitations of the studied regime (infinite width assumed in the analysis unlike the most recent results studying the NTK regime). I agree with the reviewers that comparing the NTK conditioning of ReLU and linear activations is of limited novelty. The resuld would be much stronger if it can be refined to distinguish the conditioning of two non-linear activations.

Given the current form of the paper, I regret to inform you that I am unable to recommend the acceptance of the paper for publication at ICLR. I want to emphasize that this decision should not be viewed as a discouragement. In fact, the reviewers and I believe that your work has valuable insights and, with further development and refinement, can make a meaningful impact on the field.

I encourage you to carefully address the feedback provided by the reviewers and consider resubmitting the paper. Please use the comments and suggestions in the reviews to improve and refine your work.

Best,
AC

**Additional Comments On Reviewer Discussion:**

Reviewers RSh8 and Reviewer RSh8 pointed out issues including (i) limited novelty of the conclusion that ReLU NTK is better than the linear NTK  (ii) limited scope of the framework (e.g., type of activations) (iii) limitations of the studied regime (infinite width assumed in the analysis unlike the most recent results studying the NTK regime). The authors provide a rebuttal. However, the reviewers did not find this rebuttal neither detailed enough nor convincing.

---

### Decision · Program_Chairs · 2025-01-22

Reject